# Emergence of Calabi–Yau manifolds in high-precision black-hole scattering

Mathias Driesse[1], Gustav Uhre Jakobsen[1,2], Albrecht Klemm[3,4], Gustav Mogull[1,2,5], Christoph Nega[6], Jan Plefka[1✉], Benjamin Sauer[1] & Johann Usovitsch[1]

When two massive objects (black holes, neutron stars or stars) in our universe fly past each other, their gravitational interactions deflect their trajectories[1,2]. The gravitational waves emitted in the related bound-orbit system—the binary inspiral—are now routinely detected by gravitational-wave observatories[3]. Theoretical physics needs to provide high-precision templates to make use of unprecedented sensitivity and precision of the data from upcoming gravitational-wave observatories[4]. Motivated by this challenge, several analytical and numerical techniques have been developed to approximately solve this gravitational two-body problem. Although numerical relativity is accurate[5–7], it is too time-consuming to rapidly produce large numbers of gravitational-wave templates. For this, approximate analytical results are also required[8–15]. Here we report on a new, highest-precision analytical result for the scattering angle, radiated energy and recoil of a black hole or neutron star scattering encounter at the fifth order in Newton's gravitational coupling $G$, assuming a hierarchy in the two masses. This is achieved by modifying state-of-the-art techniques for the scattering of elementary particles in colliders to this classical physics problem in our universe. Our results show that mathematical functions related to Calabi–Yau (CY) manifolds, $2n$-dimensional generalizations of tori, appear in the solution to the radiated energy in these scatterings. We anticipate that our analytical results will allow the development of a new generation of gravitational-wave models, for which the transition to the bound-state problem through analytic continuation and strong-field resummation will need to be performed.

Shortly after writing down his theory of general relativity, Einstein postulated the existence of gravitational waves[1]. Just as accelerated charges give rise to electromagnetic waves, so too do accelerated masses generate gravitational radiation. It took one hundred years to reach the technical ability to first detect gravitational waves[16] as they emerge from binary inspirals and mergers of black holes and neutron stars—the highest mass density objects in our universe. Today, more than one hundred such events have been observed by the LIGO, Virgo and KAGRA detectors[3]. The planned third generation of ground-based and space-based gravitational-wave observatories[17–19], including the recently approved LISA mission, will reach an experimental accuracy enabling unprecedented insights into gravitational, astrophysical, nuclear and fundamental physics.

To benefit from the increased sensitivity of gravitational-wave detectors, corresponding increases in precision are required in our ability to solve the gravitational two-body problem[4], described by the highly nonlinear Einstein field equations, and thus predict the gravitational waves produced in a binary encounter. Although numerical relativity, which discretizes spacetime and solves the resulting equations numerically on supercomputers, provides a good option[5–7],

it is slow and computationally expensive (a run for a single configuration can take weeks). As tens of millions of waveform templates are needed for gravitational-wave data analysis, fast approximate analytical results to the two-body problem are therefore also required. These can be generated using perturbation theory, picking one or more small parameters and solving the equations order by order in a series expansion. For the binary inspiral, this includes a slow-velocity and weak-gravitational-field expansion (post-Newtonian)[8–10], as well as the semi-analytical self-force expansion[20–22] in a small mass ratio ($m_1/m_2 \ll 1$).

Here we describe a black hole (or neutron star) scattering encounter, which—although asymptotically unbounded—also yields physical data concerning the bound two-body problem, relevant for binary inspirals[23,24]. In the scattering regime, we may advantageously make use of a weak-gravitational-field expansion, in powers of Newton's constant $G$, valid as long as the two bodies are well separated but moving at arbitrary velocities[11–15,25] (Fig. 1). The first non-trivial order for such a black-hole scattering was found in 1979 (ref. 26), namely, the sub-leading $G^2$ order[27]. Rapid progress has been made since then by synergistically porting techniques from quantum field theory (QFT)[12–14,28], the mathematical

[1]Institut für Physik, Humboldt-Universität zu Berlin, Berlin, Germany. [2]Max-Planck-Institut für Gravitationsphysik (Albert-Einstein-Institut), Max Planck Society, Potsdam, Germany. [3]Bethe Center for Theoretical Physics, Universität Bonn, Bonn, Germany. [4]Hausdorff Center for Mathematics, Universität Bonn, Bonn, Germany. [5]Centre for Theoretical Physics, Department of Physics and Astronomy, Queen Mary University of London, London, UK. [6]Physik-Department, TUM School of Natural Sciences, Technische Universität München, Garching, Germany. ✉e-mail: jan.plefka@hu-berlin.de

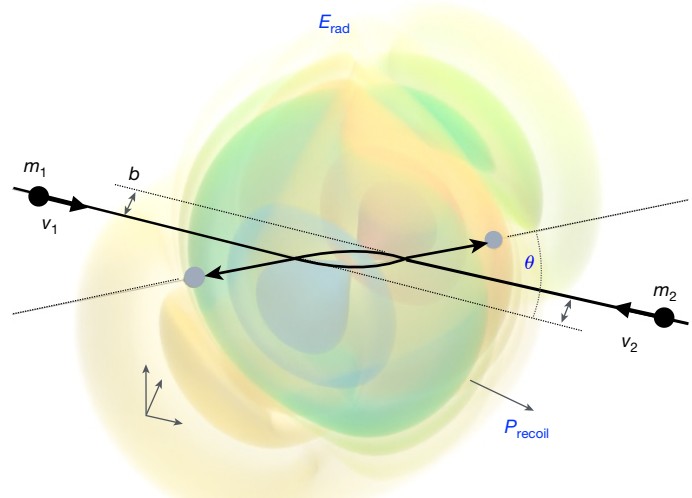

$E_{rad}$

$m_1$
$v_1$

$b$

$\theta$

$m_2$
$v_2$

$P_{recoil}$

**Fig. 1 | Gravitational two-body scattering event.** Two black holes (or neutron stars) with masses $m_i$ and incoming velocities $v_i$, impact parameter $b$ and resulting relative scattering angle $\theta$, radiated gravitational-wave energy $E_{rad}$ and recoil (shown in blue).

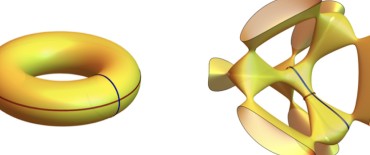

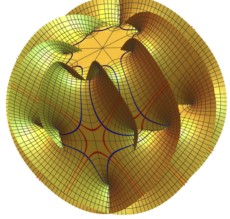

Torus ($n = 1$) K3 ($n = 2$) CY3 ($n = 3$)

**Fig. 2 | Graphical representation of the CY $n$-folds emerging in black-hole scattering.** The elliptic curve (topologically a torus) and two-dimensional projections of the K3 surface and CY3 reflecting their symmetries. Red and blue lines are (projections of) the real $n$-dimensional cycles $\Gamma_n$. The corresponding periods over the $n$-form $\Omega_n(x)$, that is, $\int_{\Gamma_n} \Omega_n(x)$, depend on the so-called modulus $x$ (related to the relative velocity $v$ of the black holes $1/\sqrt{1-(v/c)^2} = v_1 \cdot v_2/c^2 = (x+x^{-1})/2$) parametrizing the shape of CY manifolds and yield master integrals in our problem.

framework for elementary particle scattering, to this classical physics problem. The third order ($G^3$) was established in 2019 (refs. 29–32) and the fourth order ($G^4$) was completed in 2023 (refs. 33–37). At least the fifth order ($G^5$) precision will be needed to prepare for the third generation of gravitational-wave detectors[4].

Being characterized by three fundamental properties—their mass, spin and charge—black holes are, in a sense, the astrophysical equivalents of elementary particles. QFT is a highly mature subject and precise analytic predictions for particle scattering events, used at colliders, such as CERN's Large Hadron Collider, are commonplace. We benefit from this progress in gravity through the close theoretical link between hyperbolic motion (unbound scattering) and elliptic motion (bound orbits). State-of-the-art technologies for performing the multi-loop Feynman integrals involved in scattering cross-sections have enabled some remarkable predictions in elementary particle physics[38–41] and uncovered surprising connections to algebraic geometry[42–45].

The link to algebraic geometry arises through the function spaces that are needed to express the observables at growing perturbative orders. We typically find generalizations of logarithms, known as multiple polylogarithms, which are well understood. At higher orders, elliptic integrals make their appearance[46]. Geometrically, these are period integrals over the two non-trivial closed cycles of a family of elliptic curves (also known as tori) (Fig. 2). The physical parameters determine the shape of the latter. Yet, this is just the tip of the iceberg.

Recently, it has become evident[43–45,47,48] that CY manifolds emerge in generalizations of the aforementioned function spaces, which encode Feynman integrals in higher-order perturbation theory. These are complex $n$-dimensional manifolds whose metric obeys Einstein's vacuum equations in $2n$-spacetime dimensions[49]. Geometrically, these higher-dimensional CY $n$-folds represent a beautiful series of critical geometries generalizing the elliptic curve ($n = 1$) and may be thought of as $2n$-dimensional generalizations of the torus. To motivate this, consider the Legendre family of elliptic curves $Y^2 = X(X-1)(X-x)$ with $X$ and $Y$ complex variables. The one-form $\Omega_1 = dX/Y$ yields the elliptic periods $\varpi_0 = 2\int_1^\infty dX/Y$ and $\varpi_1 = -2i\int_{-\infty}^0 dX/Y$, which are expressible through standard elliptic integrals. These satisfy a second-order differential equation $(1 + 4(2x-1)\partial_x + 4x(x-1)\partial_x^2)\varpi_k = 0$ for $k = 0, 1$, known as a Picard–Fuchs equation. In turn, CY $n$-folds exhibit an $n$-form $\Omega_n$, whose periods—integrals over higher-dimensional integration cycles (Fig. 2)—generalize the elliptic integrals. These $n$-fold periods

obey Picard–Fuchs equations of order ($n + 1$). Although CY twofolds—known as K3 surfaces[50]—have a unique topology, the topological types of CY ($n > 2$)-fold are not classified but believed to be finite. CY three-folds (CY3) are of particular interest in string theory, in which they are used to curl up the six extra spacetime dimensions to arrive at the four observable spacetime dimensions[51].

Although specific higher-loop Feynman integrals are known to be expressed in terms of CY periods[43–45,47,48], physical observables tend to be much simpler than the multitude of contributing Feynman integrals. For example, the Feynman integrals occurring in black-hole scattering at orders $G^4$ (refs. 33,34) and $G^5$ (ref. 52) encode K3 periods, but these contributions strongly cancel in the physical observable within the conservative sector at $G^5$ (ref. 53). Similar intriguing cancellations occur in QFT computations[54,55]. Furthermore, CY $n$-fold periods have a transcendentality degree[45,56] increasing with their dimension $n$. This leads to the important question of what classes of transcendental functions appear in physical observables in perturbation theory. Before our work, no physical observables were known that feature CY $n$-fold periods for $n \geq 3$. We expect that our findings and the methods described below will have substantial implications for high-precision predictions in particle physics as well.

In this article, we report on a new landmark result of the QFT-based classical general relativity programme by providing complete scattering observables of a binary black hole (or neutron star) encounter up to the fifth order in the weak-field expansion ($G^5$) and sub-leading order in the symmetric mass ratio $\nu = m_1 m_2/(m_1 + m_2)^2$. This encounter is depicted in Fig. 1 and involves two black holes scattering with a deflection angle $\theta$ and radiating gravitational waves with total energy $E_{rad}$. We describe the black holes as point particles, an approximation valid as long as their separation $b$ is large compared with their intrinsic sizes, that is, their Schwarzschild radii $2Gm_i/c^2$—the weak-gravitational-field region. Consequently, the $G$ expansion is really an expansion in the dimensionless quantity $GM/bc^2$ with total mass $M = m_1 + m_2$. The two scattering observables $\theta$ and $E_{rad}$, the latter depending on CY3 periods, can be used to calibrate gravitational-waveform models.

The gravitationally interacting two-body system is governed by an action consisting of two worldlines, coupled to the gravitational Einstein–Hilbert term:

$$S = -m_1 c \int ds_1 - m_2 c \int ds_2 - \frac{c^3}{16\pi G} \int d^4x \sqrt{-g} R[g]. \quad (1)$$

Variation of this action gives rise to the Einstein and geodesic equations. To explain our notation, the proper time intervals $ds_i = \sqrt{g_{\mu\nu} \dot{x}_i^\mu \dot{x}_i^\nu}\, d\tau$ give rise to the followed trajectories $x_i^\mu(\tau)$ ($\mu = 0, 1, 2, 3$) of the $i$th black hole, parametrized by a time parameter $\tau$ (a dot symbolizes a

$\tau$ derivative). The spacetime metric $g_{\mu\nu}(x)$ yields the curvature scalar $R[g]$ and $g = \det(g_{\mu\nu})$.

We calculate the change in four-momentum of each body over the course of scattering, $\Delta p_i^\mu$, known as the impulse. With the momentum of each body given by $p_i^\mu = m_i \dot{x}_i^\mu$, the impulse is simply the difference between the momentum at late and early times:

$$\Delta p_i^\mu = p_i^\mu(\tau \to +\infty) - p_i^\mu(\tau \to -\infty)$$
$$= G\Delta p_i^{(1)\mu} + G^2\Delta p_i^{(2)\mu} + G^3\Delta p_i^{(3)\mu} + G^4\Delta p_i^{(4)\mu} + G^5\Delta p_i^{(5)\mu} + \cdots. \quad (2)$$

The initial momentum of each black hole is given by its mass times initial velocity, $p_i^\mu(\tau \to -\infty) = m_i v_i^\mu$. Working in a weak-gravitational-field region, we have series-expanded order by order in Newton's constant. With results up to $G^4$ already determined[33,35,36], and the conservative (non-radiating) part of $G^5$ derived by some of the present authors[53], here we extract the subleading-in-$v$ $G^5$ component from which we will also derive the scattering angle $\theta$ and radiated energy flux $E_{\mathrm{rad}}$. Note that $v$ tends to zero for $m_1 \ll m_2$ and vice versa.

Our calculation is performed using Worldline Quantum Field Theory (WQFT)[11,57], in which a Worldline Effective Field Theory action is used to represent the black holes as point particles[25,28]. This allows us to reinterpret this classical physics problem as one of drawing and calculating perturbative Feynman diagrams (Extended Data Fig. 1). The main benefit of WQFT for classical physics computations is a clean separation between classical and quantum effects. In this language, gravitons (wavy lines) and deflection modes (solid lines) are the quantized excitations of the metric $g_{\mu\nu}$ and trajectories $x_i^\mu$. The momenta and energies of these particles are unfixed and must be integrated over. The key principle being exploited here is that tree-level one-point functions, given by a sum of diagrams with a single outgoing line and no internally closed loops, solve the classical equations of motion[58]. We recursively generated the graphs to be computed at the fifth order in $G$, yielding a total of 426 diagrams. These diagrams directly translate to mathematical expressions, Feynman integrals, by way of Feynman rules derived from the action in equation (1) (Extended Data Fig. 2).

The resulting Feynman integrals are a staple of perturbative QFT. Individual Feynman integrals, which may diverge in four spacetime dimensions, are treated by working in $D$ dimensions so that divergences appear as $(D-4)^{-1}$ poles. Finiteness of our results in the limit $D \to 4$, that is, the cancellation of all intermediate divergences, then provides a useful consistency check. Our calculation of the impulse calls for the evaluation of millions of Feynman integrals, which may have at most 13 propagators of the kinds seen in Extended Data Fig. 2. To evaluate them, we generate linear integration-by-parts (IBP) identities, which reduces the problem to one solving a large system of linear equations. The task was nevertheless enormous and consumed around 300,000 core hours on high-performance computing clusters.

Our task is ultimately to determine expressions for a basis of 236 + 232 master integrals, which split under parity ($v_i^\mu \to -v_i^\mu$) into two distinct sectors. From these, all other integrals may be expressed as linear combinations using the IBP identities. To do so, we exploit the integrals' non-trivial dependence on only a single variable $x$: it derives from the relativistic boost factor $\gamma = 1/\sqrt{1-(v/c)^2} = v_1 \cdot v_2/c^2$ for the initial relative velocity $v$ of the two black holes, through $\gamma = (x+x^{-1})/2$. Rather than attempt a direct integration, we may therefore set up two systems of differential equations in $x$ (even and odd parity) as:

$$\frac{\mathrm{d}}{\mathrm{d}x}\mathbf{I}(x,D) = \hat{M}(x,D)\mathbf{I}(x,D). \quad (3)$$

The integrals to be computed are grouped into vectors $\mathbf{I}$ and the matrices $\hat{M}$ take a lower block triangular form (Fig. 3). To obtain this system, derivatives of the master integrals with respect to $x$ are re-expressed as linear combinations of the masters themselves using the IBP identities. We solve the system order by order in a series expansion close to $D = 4$, with higher-order terms given by repeated integrals (with respect to $x$) of lower-order terms. Boundary conditions on the integrals are fixed in the non-relativistic (low-velocity) limit $x \to 1$.

Repeated integrations with respect to the kinematic parameter $x$ produce the mathematical functions $\mathcal{I}$ appearing in our final result for the impulse:

$$\mathcal{I}(\phi_1, \phi_2, ..., \phi_n; x) := \int_1^x \mathrm{d}x'\phi_1(x')\mathcal{I}(\phi_2, ..., \phi_n; x'), \quad (4)$$

with the base case $\mathcal{I}(; x) = 1$. The nature of the integration kernels $\phi_i$ determines the types of function and are associated with underlying geometries. In the simplest case, the kernels $\phi_i$ are rational functions with single poles, for example, $x^{-1}$, $(x+1)^{-1}$ or $(x-1)^{-1}$, and iterated integrations produce the function class of multiple polylogarithms—including the ordinary logarithm $\mathcal{I}(x^{-1}; x) = \log x$. Geometrically, we can interpret these integration kernels as periods of a zero-dimensional CY space, given by two points on a sphere. In more complicated scenarios, usually related to higher-loop computations, the $\phi_i$ are connected to periods of higher-dimensional algebraic varieties. A key challenge is to understand the kernels and the associated class of iterated integrals occurring in a physical problem. In a $G^4$ calculation of the impulse, squares of elliptic integrals arise, which are geometrically interpreted as periods of a one-parameter K3 surface (Fig. 2). In the odd-parity sector of integrals at the present fifth order in $G$, the kernels also depend on CY3 periods and we express physical quantities in terms of the corresponding class of iterated functions.

To see the origin of the CY3 periods, and to clarify their precise nature, we examine the differential equation matrix $\hat{M}(x,D)$ (see Fig. 3) in the limit $D \to 4$. The diagonal blocks of this matrix are associated with specific Feynman graphs appearing in the impulse, of which the CY3 geometry is isolated to a single $4 \times 4$ diagonal sub-block. We can decouple these four first-order differential equations such that we obtain a single fourth-order differential equation, which is the Picard–Fuchs equation of the CY threefold:

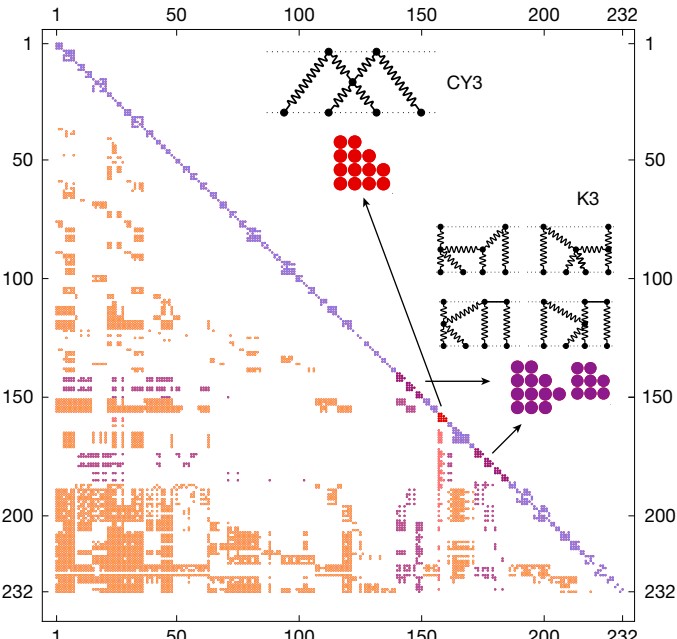

**Fig. 3 | Non-zero entries of the odd-parity 232 × 232 differential equation matrix $\hat{M}(x,D)$.** The blocks on the diagonals determine the function spaces of the multiple sub-sectors. The unmagnified diagonal sectors give rise to multiple polylogarithms.

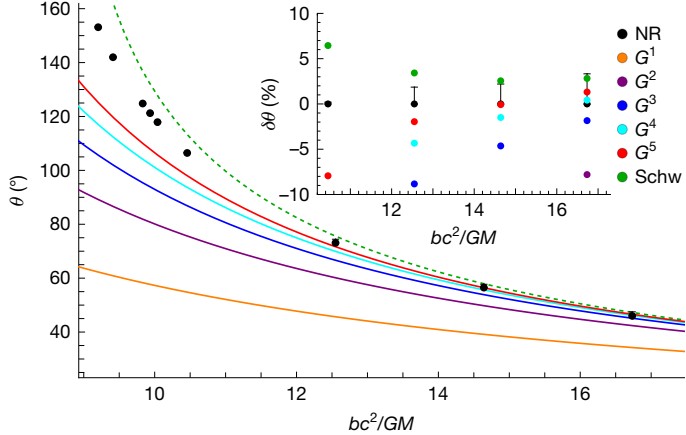

**Fig. 4 | The scattering angle θ.** Scattering angle $\theta$ is plotted as a function of the impact parameter in units of the Schwarzschild radius, $bc^2/GM$, up to order $G^5$ for an equal-mass scenario with initial relative velocity $v = 0.5125c$. The black dots are existing numerical relativity (NR) simulations[60]. The $G^5$ curve follows from equation (6) (excluding the unknown $v^2\theta^{(5,2)}$ contribution). The dashed line is the exact in $G$ ($v = 0$) probe limit result for geodesic motion in a Schwarzschild background. The inset plot depicts the relative differences to the numerical relativity data. Larger values of $bc^2/GM$ correspond to the perturbative regime. We find agreement with NR within the error for $bc^2/GM > 12.5$. The monotonically falling corrections to the consecutive $G^n$ orders yield an intrinsic error estimate of our $G^5$ results: they are more precise than the NR data for $bc^2/GM > 14$.

$$\left[\left(x\frac{d}{dx} - 1\right)^4 - x^4\left(x\frac{d}{dx} + 1\right)^4\right]\varpi(x) = 0. \tag{5}$$

The latter is solved by the four CY threefold periods $\varpi_k(x) = \int_{\Gamma_3^k}\Omega_3(x)$, in which the three-form $\Omega_3(x)$ is integrated over the real three-dimensional cycles $\Gamma_3^k$, $k = 0, 1, 2, 3$. For an algebraic definition of the CY family together with an explicit expression of $\Omega_3$, we refer to refs. 52,59. These integrals appear within the integration kernels $\phi_i$ of the iterated integrals (equation (4)).

Our final expression for the fifth-order impulse is involved and described in Methods, in which we also elaborate on the function space. From the impulse, we can derive the scattering angle $\theta$, which measures the angle of deflection between the ingoing and outgoing momenta in the initial centre-of-mass inertial frame (Fig. 1). As the system dissipates energy, it recoils, and so the initial frame choice is not preserved over the course of a scattering event. Like the impulse (equation (2)), the scattering angle is expanded in the weak-field limit with the $G^n$ component, denoted $\theta^{(n)}$. These components are also expanded in powers of the symmetric mass ratio $v$ and at order $G^5$, we have

$$\theta^{(5)} = \frac{M^5\Gamma}{b^5c^{10}}\left(\theta^{(5,0)} + v\theta^{(5,1)} + v^2\theta^{(5,2)} + v^3\Gamma^{-2}\theta^{(5,3)}\right), \tag{6}$$

with $M$ and $\Gamma = E/M$ the total mass and mass-rescaled energy of the initial system, respectively. A central result of our work is the computation of all contributions except for $\theta^{(5,2)}$. The function space of the angle $\theta^{(5)}$ arises from integrals only in the even-parity sector and is simpler than that of the complete impulse. We compare our result with available numerical relativity simulations[60] in Fig. 4.

Our other main result is the total radiated energy and momentum from the system over the course of the scattering. Using the principle of four-dimensional momentum conservation, which includes conservation of energy, the total loss of momentum through gravitational-wave emission must balance the change in momenta of the two individual black holes (or neutron stars):

$$P_{rad}^\mu = -\Delta p_1^{\ \mu} - \Delta p_2^{\ \mu}. \tag{7}$$

The impulse of the second black hole, $\Delta p_2^{\ \mu}$, can straightforwardly be inferred from that of the first using symmetry. The radiated energy, then, is given simply by the zeroth component of the radiated four-momentum in the centre-of-mass frame $E_{rad} = P_{rad}^0 = -\Delta p_1^{\ 0} - \Delta p_2^{\ 0}$, whereas the recoil $\mathbf{P}_{recoil}$ derives from its spatial components. Unlike the scattering angle, it includes integrals from the odd-parity sector and so contains CY periods. These terms contribute to the repeated backscattering of radiative gravitons off the potential background—known as the 'tail-of-tail' effect.

Summarizing, in this work, we have extended the state of the art of the gravitational two-body problem to a new perturbative order ($G^5$) to the sub-leading mass ratio level $v$. Our analytical findings require the use of a new class of functions, CY threefold periods, in the radiative sector. These methodological advances will also benefit particle phenomenology, in which CY periods appear in higher-loop-order diagrams[43–45,47,48]. By comparing with numerical relativity data, we demonstrated percent-level agreement in the perturbative domain. These results provide input data for high-precision waveform models using effective-one-body resummation techniques[24,60–62] that can now be developed. For the comparable-mass case, we foresee the need to also incorporate next-to-next-to-leading-order mass ratio ($v^2$) contributions, in which new CY threefolds are expected to make their appearance[52]. This we leave for future studies.

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

## Methods

### Integrand generation and integral family

We use the WQFT formalism[11,57,63] that quantizes the worldline deflections $z_i^{\mu}(\tau)$ and graviton field $h_{\mu\nu}(x)$ arising in the background field expansions $x_i^{\mu}(\tau) = b_i^{\mu} + v_i^{\mu}\tau + z_i^{\mu}(\tau)$ and $g_{\mu\nu} = \eta_{\mu\nu} + \sqrt{32\pi G}\, h_{\mu\nu}$, respectively (now setting $c = 1$). In the gravitational sector, we use a nonlinearly extended de Donder gauge that simplifies the three-graviton and four-graviton vertices (see Supplementary Information). The worldline actions (equation (1)) are improved by making use of the proper time gauge $\dot{x}_i^2 = 1$ for the $i$th black hole:

$$S_i = -\frac{m_i}{2} \int d\tau\, g_{\mu\nu}[x_i(\tau)]\, \dot{x}_i^{\mu}(\tau)\dot{x}_i^{\nu}(\tau). \tag{8}$$

This ensures a linear coupling to the graviton $h_{\mu\nu}$. At the present four-loop ($G^5$ or 5PM) order, we require up to six-graviton vertices that derive from the Einstein–Hilbert action plus gauge-fixing term—taken in $D = 4 - 2\epsilon$ dimensions. We also require the single-graviton emission plus (0,..., 5)-deflection vertices derived from equation (8). We provide the explicit vertices and graviton gauge-fixing function in a Zenodo repository submission[64] accompanying this article; an analytic expression for the $n$-deflection worldline vertex was given in refs. 11,63.

The full 5PM integrand is generated using the Berends–Giele-type recursion relation discussed in ref. 65 and sorted into five self-force (SF) sectors according to their scaling with the masses $m_1$ and $m_2$:

$$\Delta p_1^{(5)\mu} = m_1 m_2 \big( m_2^4 \Delta p_{0SF}^{(5)\mu} + m_1 m_2^3 \Delta p_{1SF}^{(5)\mu} \\ + m_1^2 m_2^2 \Delta p_{2SF}^{(5)\mu} + m_1^3 m_2 \Delta p_{1SF}^{(5)\mu} + m_1^4 \Delta p_{0SF}^{(5)\mu} \big). \tag{9}$$

The powers of the masses follow from the number of times a worldline is touched in a given graph. Here we compute the sub-leading self-force (1SF) contributions $\Delta p_{1SF}^{(5)\mu}$ and $\Delta p_{1SF}^{(5)\mu}$, as well as reproducing the 0SF contributions $\Delta p_{0SF}^{(5)\mu}$ and $\Delta p_{0SF}^{(5)\mu}$ that follow from the geodesic motion in a Schwarzschild background. The resulting integrand is reduced to scalar integrals by means of tensor reduction and 'planarized' using partial fraction (eikonal) identities as described in ref. 53. In summary, all integrals are mapped to the 5PM-1SF planar family $\int_{\ell} := \int d^D\ell/(2\pi)^D, \bar{\delta}(x) := 2\pi\delta(x)$

$$\mathcal{I}_{\{n\}}^{\{\sigma\}} = \int_{\ell_1 \cdots \ell_4} \frac{\bar{\delta}^{(\bar{n}_1-1)}(\ell_1 \cdot v_1) \prod_{i=2}^{L} \bar{\delta}^{(\bar{n}_i-1)}(\ell_i \cdot v_2)}{\prod_{i=1}^{4} D_i^{n_i}(\sigma_i) \prod_{I<J} D_{IJ}^{n_{IJ}}}, \tag{10a}$$

in which $\{\sigma\}$ and $\{n\}$ denote causal $i0^+$ prescriptions and integer powers of propagators, respectively. The four worldline propagators $D_i(\sigma_i)$ appearing are ($i = 2, 3, 4$)

$$D_1 = \ell_1 \cdot v_2 + \sigma_1 i0^+, \quad D_i = \ell_i \cdot v_1 + \sigma_i i0^+ \tag{10b}$$

and the 14 gravitons propagators $D_{IJ}$ with $I = (0, 1, i, q)$ are

$$D_{1j} = (\ell_1 - \ell_j)^2 + \sigma_{4+j}\mathrm{sign}(\ell_1^0 - \ell_j^0)i0^+, \quad D_{q1} = (\ell_1 + q)^2 \\ D_{ij} = (\ell_i - \ell_j)^2, \quad D_{qi} = (\ell_i + q)^2, \quad D_{01} = \ell_1^2, \quad D_{0i} = \ell_i^2. \tag{10c}$$

There are at most three bulk graviton propagators $D_{1i}$ that may go on-shell at 5PM order.

### IBP reduction

IBP identities[66–68] are used to reduce to master integrals. We use a future release of KIRA 3.0 (refs. 69,70) adapted to our needs that uses the FireFly[71,72] library for reconstructing rational functions through finite-field sampling. We have 45 top-level sectors in the 5PM-1SF family that have been described in ref. 53. The integrals encountered in the planar family (equations (10a)–(10c)) have propagator powers in the range $n_{i/IJ} \in [-9, 8]$. The strategies applied to reduce the runtime of the IBP reduction are comparable with the conservative case[53]. The final set of needed IBP replacement rules to master integrals generated comprises about 30 GB of data and can be made available on request.

### Differential equations and function space

The method of differential equations[73–75] is used, in which the matrices in equation (3) depend on the parameters $x$ and the dimensional regulator $\epsilon = (4 - D)/2$. We take the physical limit $\epsilon \to 0$ to compute our observables. Therefore, we need the solutions of the integrals expanded in $\epsilon$. To systematically compute this expansion, we transform equation (3) into canonical form[75] such that the $\epsilon$ dependence is factored out of the differential equation matrix:

$$\frac{d}{dx}\mathbf{J}(x, \epsilon) = \epsilon\hat{A}(x)\mathbf{J}(x, \epsilon), \tag{11}$$

with $\mathbf{J}(x, \epsilon) = \hat{T}(x, \epsilon)\mathbf{I}(x, \epsilon)$ and $\epsilon\hat{A}(x) = (\hat{T}(x, \epsilon)\hat{M}(x, \epsilon) + d\hat{T}(x, \epsilon)/dx)T(x, \epsilon)^{-1}$. The solution is then a path-ordered matrix exponential:

$$\mathbf{J} = \mathcal{P}\exp\left[\epsilon\int_1^x dx'\hat{A}(x')\right]\mathbf{j}, \tag{12}$$

in which $\mathbf{j}$ encodes the boundary values of our integrals at $x = 1$.

We take a bottom-up approach to determine the required transformation $\hat{T}(x, \epsilon)$. For this, we sort our integrals into groups sharing the same set of propagators. These so-called sectors are ordered from lower (fewer propagators) to higher (more propagators), resulting in the block diagonal matrix in Fig. 3. We begin by $\epsilon$-factorizing the lower sectors and then move on to the higher sectors. First, we transform the diagonal blocks, which are identified with the maximal cuts[76], into $\epsilon$-form and then proceed to the off-diagonal contributions. The IBP reductions were fully completed for all integrals before the diagonal blocks were identified. Particularly for handling sectors coupled to the CY3 diagonal sector, it is important to choose a good initial basis of integrals such that the relevant couplings are as simple as possible. As we proceed to canonicalize, we adapt and improve our choice of initial basis accordingly.

The simplest diagonal blocks to canonicalize are those containing only multiple polylogarithms[77–79], depicted in lilac in Fig. 3. The algorithm CANONICA[80] finds the necessary transformation to $\epsilon$-form for these blocks by making a suitable ansatz. It is noteworthy that the complexity of this transformation, as well as the runtime, depends highly on the choice of initial integrals. In general, we pick our initial basis integrals so that the regulator $\epsilon$ does not appear non-trivially in the denominators of the differential equation (3).

Diagonal blocks containing a K3 surface, depicted in purple in Fig. 3, are handled using INITIAL[81]. The INITIAL algorithm requires a pure seed integral to construct an $\epsilon$-factorized differential equation through an ansatz tailored to the specific seed integral. Note that, in this context, a pure integral is given as a linear combination of iterated integrals (equation (4)) having no non-trivial pre-factors in $x$. For non-pure integrals, these pre-factors are non-trivial functions and are known as leading singularities. For each K3 sector, we find an appropriate seed integral by analysing the diagonal block at $\epsilon = 0$. We decouple each diagonal K3 block by switching to a derivative basis: $\mathbf{I}_{K3} = (I_1, I_2, I_3) \to (I_1, I_1', I_1'')$ or $\mathbf{I}_{K3} = (I_1, I_2, I_3, I_4) \to (I_1, I_1', I_1'', I_4)$, depending on the size of the block, in which $I_i$ are the master integrals of this sector—$I_4$ is chosen so that it decouples from $I_1$ and its derivatives as much as possible. The choice of $I_1$ ensures that its third-order differential (Picard–Fuchs) equation ($\hat{\theta} = x\frac{d}{dx}$),

$$[\hat{\theta}^3 - 2x^2(2 + 4\hat{\theta} + 3\hat{\theta}^2 + \hat{\theta}^3) + x^4(2 + \hat{\theta})^3]I_1|_{\epsilon=0} = 0, \tag{13}$$

has the explicit solution $I_1|_{\epsilon=0} \propto \varpi_{K3} = \left(\frac{2}{\pi}\right)^2 K^2(1 - x^2)$, that is, it is proportional to a K3 period. Unlike the polylogarithmic case, this

third-order differential equation is not factorizable into first-order equations. Using the normalized integral $I_1/\varpi_{K3}$ as the seed, INITIAL may then construct an $\epsilon$-form for the diagonal parts of our K3 sectors (similar to the 4PM case).

To canonicalize the single diagonal CY3 block, depicted in red in Fig. 3, we follow the discussion in ref. 52. Similar to K3, we first pick a suitable starting integral and make a basis change $\mathbf{I}_{CY3} = (I_1, I_2, I_3, I_4) \rightarrow (I_1, I_1', I_1'', I_1''')$. The starting integral $I_1$ is chosen so that, when $\epsilon = 0$, it satisfies the Picard–Fuchs equation (5), which is a hypergeometric system. This implies that the periods of our CY3 geometry are given in terms of hypergeometric functions, for example, $x_4F_3\left[\frac{1}{2}, \frac{1}{2}, \frac{1}{2}, \frac{1}{2}; 1, 1, 1; x^4\right]$. Moreover, this also gives rise to intriguing arithmetic properties discussed in ref. 59. From this equation and its four fundamental solutions $\varpi_0, \ldots, \varpi_3$, which we collect in the Wronskian matrix $W(x) = (\partial^j \varpi_i)$ with $0 \leq i, j \leq 3$, we construct in several steps the rotation matrix into $\epsilon$-form. This approach was invented in ref. 82 and further developed in ref. 83, and, in the K3 case, it is equivalent to the INITIAL algorithm. The process involves the following three steps:

1. We split the Wronskian matrix into a semi-simple and unipotent part $\hat{W} = \hat{W}^{ss} \times \hat{W}^u$. Naively, we can understand this splitting as a decomposition of the maximal cuts of the CY3 sector into their leading singularities and pure parts. The unipotent part is named after the unipotent differential equation it fulfils:

$$(\mathrm{d} - A^u(x))\hat{W}^u(x) = 0, \tag{14}$$

in which $A^u(x)$ is nilpotent. The matrix $A^u(x)$ can generally be written in terms of invariants, known as $Y$-invariants, of the CY variety and was explicitly given for our CY3 in ref. 52.

2. We rotate our integrals with the inverse of $W^{ss}$, which strips them of their leading singularities. (The analogous operation for K3 is normalizing $I_1$ with the K3 period $\varpi_{K3}$, leaving only the pure part.) To parametrize all degrees of freedom in $W^{ss}$ of a CY3, we need the holomorphic solution $\varpi_0$, an extra function

$$\alpha_1 = \frac{\varpi_0^2}{x(\varpi_0\varpi_1' - \varpi_0'\varpi_1)}, \tag{15}$$

called the structure series, and their derivatives. The appearance of $\alpha_1$ is new compared with the K3 case and shows the increased complexity in structure of a CY3. In ref. 84, the structure series $\alpha_1$ (and more generally the $Y$-invariants) are generally defined and used to construct a normal form of a CY differential equation; more specifically, for our CY3, they were derived in ref. 52. For $\epsilon$-factorizing our differential equations, it is important that this normal form of the CY differential equation is in a factorized form with respect to its derivatives. To eliminate all redundancies in this step, we must use Griffiths transversality—an essential property of CY geometries that yields quadratic relations between their periods—to simplify the form of $\hat{W}^{ss}$.

3. After completing these steps and appropriately rescaling in $\epsilon$ to arrange the weights of the integrals, the diagonal block of our CY3 looks like:

$$\frac{\mathrm{d}}{\mathrm{d}x}\tilde{\mathbf{I}}_{CY3} = \sum_{i=-2}^{1} \epsilon^i \hat{\mathbf{M}}_{CY3}^i(x)\tilde{\mathbf{I}}_{CY3}, \text{ with}$$

$$\tilde{\mathbf{I}}_{CY3} = T_{\epsilon-\text{scalings}}(\hat{W}^{ss})^{-1}\mathbf{I}_{CY3}. \tag{16}$$

We find an $\epsilon$-form by acting with a suitable set of transformation matrices on $\tilde{\mathbf{I}}_{CY3}$, working order by order in $\epsilon$ starting from $\epsilon^{-2}$. The process requires us to introduce four new functions $G_i(x)$ ($i = 1, \ldots, 4$), which obey a first-order differential equation containing $\varpi_0$, $\alpha_1$, their derivatives and $G_j(x)$ functions with $j < i$. For example,

$$G_1'(x) = -\frac{96x(x^4+1)\varpi_0(x)^2}{(x-1)^2(x+1)^2(x^2+1^2)\alpha_1(x)}. \tag{17}$$

Because of this structure, the functions $G_i(x)$ are all expressible as iterated integrals of CY periods and associated functions. These functions were previously introduced in terms of a different variable in ref. 52 and are listed for our conventions in the Supplementary Information of this article.

We now have the $\epsilon$-form of the diagonal part of the CY3 sector and, thus, a canonical form of all diagonal blocks. We refer to this intermediate basis, in which all diagonal blocks are in $\epsilon$-form, as $\mathfrak{J}$. The next stage in canonicalization involves tackling the off-diagonal blocks. To do so, we distinguish between off-diagonal blocks coupled to the CY3 sector, which require special care, and the rest.

We have developed our own algorithm to transform the off-diagonal entries of our differential equation that do not couple to the CY3 sector but can depend on K3 functions. This algorithm uses FINITEFLOW[85] and MultivariateApart[86] and provides suitable ansätze also including elliptic contributions for the required transformations. It is similar to algorithms used for polylogarithmic off-diagonals, such as those found in CANONICA or Libra[87].

For sectors polylogarithmic or K3 on their diagonal blocks, yet also coupled to the CY3 sector, a good initial basis of integrals is essential to minimize these couplings. One possibility to identify such candidates is to perform an integrand analysis, usually done in the Baikov representation[88,89] of the integrals. However, we instead found it simpler to use the diagonals themselves to derive constraints on the initial choice of integrals, expanding on the ideas of ref. 90. Having now found canonical masters on the diagonals, that is, the maximal cuts, our strategy is to choose initial candidate integrals that are related as closely as possible to these canonical masters within their respective diagonal blocks. More precisely, we search for candidates that, on their diagonal blocks, are given by a linear combination of the canonical maximal cut integrals and overall functions of $\epsilon$ and $x$:

$$I_i^{\text{candidate}} = f(\epsilon)g(x)\sum_k c_k \mathfrak{J}_k, \tag{18}$$

in which the $c_k$ are constant numbers.

We expect that such a 'good' choice of candidate integrals only requires minimal corrections to form a canonical basis. For certain sectors that are polylogarithmic on their corresponding diagonal block, we need to enlarge these types of constraint by combining different polylogarithmic sectors and requiring equation (18) to hold beyond a single diagonal block. Thus, we obtain further conditions resulting from off-diagonal couplings between separate polylogarithmic blocks. In some instances, we can also relax the condition in equation (18) by considering the $c_k(x)$ as functions of $x$ and still find easy transformations. The use of IBPs makes this procedure efficient and allows us to find all transformations for the coupling to the CY3 manually, proceeding similarly as for the diagonal of the CY3 sector. We build successive transformations, removing iteratively all nonlinear-in-$\epsilon$ contributions, starting with the highest negative power of $\epsilon$. By doing so, for some integrals, we need to introduce 16 new functions $G_5(x), \ldots, G_{20}(x)$, which again satisfy first-order differential equations listed in the Supplementary Information. More specifically, for the mixings between K3 and CY3 sectors, we introduce new functions whose first-order differential equations contain the periods of both the K3 and the CY3. For example,

$$G_8'(x) = \frac{\varpi_{K3}(x)G_3(x)\varpi_0(x)\alpha_1'(x)}{\alpha_1(x)^2}. \tag{19}$$

This concludes the canonicalization process of the whole differential equation system.

Having converted our matrix into its canonical form, $\hat{A}(x)$ provides all integration kernels needed to express the master integrals as iterated integrals (equation (4)). We selected a set of linearly independent kernels by examining their small velocity expansion. Our observables consist of iterated integrals that include K3, CY3 and mixed integration kernels, functions from the rotation matrix $\hat{T}(x, \epsilon)$ and algebraic functions from the decomposition in terms of initial master integrals. We need at most four-times-iterated integrals; all multiple polylogarithms are constructed from the kernels $\left\{\frac{1}{x}, \frac{1}{1 \pm x}, \frac{x}{1+x^2}\right\}$ and have a maximum transcendental weight of 3. Let us also note that the K3 and CY3 periods occurring above are related to those of the Legendre curve by a symmetric and a Hadamard product, respectively[52,59].

## Boundary fixing

A complete solution to the differential equation (3) requires the determination of integration constants in the form of boundary integrals, that is, master integrals in the static limit $x \to 1$ ($v \to 0$), which are functions only of $\epsilon$. As the integration and $x \to 1$ limit do not commute, we use the method of regions[91–93] to isolate contributions with definite ($\epsilon$-dependent) scalings in the velocity $v$ and series-expand integrals at the level of the integrand. These so-called regions are associated with different velocity scalings of the bulk graviton momenta $\ell_i$, which can be either potential (P) or radiative (R):

$$\ell_i^{\mathrm{P}} = (\ell_i^0, \ell_i) \approx (v, 1), \quad \ell_i^{\mathrm{R}} = (\ell_i^0, \ell_i) \approx (v, v). \tag{20}$$

There are three propagators $\{D_{12}, D_{13}, D_{14}\}$ that may enter both regions; the rest are kinematically restricted to P (including the velocity-suppressed P: $(v^2, v)$) by the presence of energy-conserving delta functions $\delta(\ell_i \cdot v_j)$. We thus denote the four possible regions as (PPP), (PPR), (PRR) and (RRR). We needed to evaluate 14 + 14 (even + odd) boundary integrals in the (PPR), 5 + 5 in the (PRR) and 4 + 4 in the (RRR) sectors, as well as the 28 + 18 (PPP) boundary integrals that were already determined in the conservative case[53] (here there is no distinction between Feynman and retarded bulk propagators). We perform all integrals analytically and check them numerically using pySecDec[94–99] and AMPred[100–103].

All new boundary integrals for us to evaluate, as compared with ref. 53, contain radiative graviton propagators. Our main strategy is to perform them by means of Schwinger parametrization, in which we also benefit by explicitly integrating out loops involving only gravitons, leaving lower-loop integrals. In doing so, we evaluate at most two-loop integrals in all but two cases. These two cases are genuine three-loop integrals for which we did not manage the integration over Schwinger parameters and therefore moved over to the time domain to establish identities at the level of the series coefficients. In general, our integrations yield generalized hypergeometric functions $_pF_q$, which can be series-expanded in $\epsilon$ by numerically expanding them to high precision and reconstructing analytic expressions using an ansatz and an integer relation algorithm. Expressions for all boundary integrals, and our methodology for deriving them, are elaborated in the Supplementary Information.

## Results

Our main results are full expressions—including dissipation—for the 1SF and $\overline{\mathrm{ISF}}$ parts of the momentum impulse, $\Delta p_{\mathrm{1SF}}^{(5)\mu}$ and $\Delta p_{\overline{\mathrm{1SF}}}^{(5)\mu}$ (equation (9)). They are expanded on basis vectors:

$$\Delta p_{\mathrm{1SF}}^{(5)\mu} = \frac{1}{b^5}\left(\hat{b}^\mu c_b(\gamma) + \check{v}_2^\mu c_v(\gamma) + \check{v}_1^\mu c_v'(\gamma)\right), \tag{21a}$$

$$\Delta p_{\overline{\mathrm{1SF}}}^{(5)\mu} = \frac{1}{b^5}\left(\hat{b}^\mu \bar{c}_b(\gamma) + \check{v}_2^\mu \bar{c}_v(\gamma) + \check{v}_1^\mu \bar{c}_v'(\gamma)\right), \tag{21b}$$

also pulling out an overall factor of the impact parameter $b$. This decomposition constitutes a split into parts originating from integrals of

even and odd parity in $v_i^\mu$. The basis vectors are the impact parameter $\hat{b}^\mu = (b_2^\mu - b_1^\mu)/b$ and dual velocity vectors $\check{v}_1^\mu = (\gamma v_2^\mu - v_1^\mu)/(\gamma^2 - 1)$ and $\check{v}_2^\mu = (\gamma v_1^\mu - v_2^\mu)/(\gamma^2 - 1)$. The main coefficients of interest here are $\{c_b, \bar{c}_b\}$ and $\{c_v, \bar{c}_v\}$, because the set $\{c_v', \bar{c}_v'\}$ being determined by lower-PM results from using preservation of mass $p_1^2 = (p_1 + \Delta p_1)^2$. The coefficients are further decomposed into those with an even or odd number of radiative gravitons in the boundary fixing:

$$c_w(\gamma) = c_{w,\mathrm{even}}(\gamma) + c_{w,\mathrm{odd}}(\gamma), \tag{22a}$$

$$\bar{c}_w(\gamma) = \bar{c}_{w,\mathrm{even}}(\gamma) + \bar{c}_{w,\mathrm{odd}}(\gamma), \tag{22b}$$

with $w \in \{b, v\}$. The even part is defined from (PPP) + (PRR) and the odd part from (PPR) + (RRR). The two parts have distinctive parities under flipping the sign of the relative velocity $v \to -v$. Even and odd sectors then give rise to integer and half-integer post-Newtonian (PN) orders, respectively.

The 1SF and $\overline{\mathrm{ISF}}$ parts of the impulse therefore each consist of four non-trivial coefficients, labelled by $b$ or $v$, each having an even or odd number of radiative gravitons R. We expand each of these in sets of basis functions $F(\gamma)$ with coefficient functions $d(\gamma)$, being polynomial up to factors of $(\gamma^2 - 1)$,

$$c_{w,z}(\gamma) = \sum_\alpha d_{w,z}^{(\alpha)}(\gamma) F_{w,z}^{(\alpha)}(\gamma)$$
$$+ \sum_\alpha d_{w,z}^{(\alpha,\mathrm{tail})}(\gamma) F_{w,z}^{(\alpha,\mathrm{tail})}(\gamma) \log(\gamma - 1), \tag{23}$$

barred coefficients being expanded in the same way. Here $z \in \{\mathrm{even, odd}\}$ counts the parity of radiative gravitons. The second line with a logarithm produced from the cancellation of $1/\epsilon$ poles between the different boundary regions is associated with tails. It is relevant for all coefficients except $c_{b,\mathrm{odd}}$, $\bar{c}_{b,\mathrm{odd}}$ and $c_{v,\mathrm{even}}$. The $b$-type basis functions are all multiple polylogarithms of maximum weight three and, thus, relatively simple. By contrast, the $v$-type basis functions are much more complex. They generally have the structure of equation (4) with kernels depending on the CY periods. All basis functions and polynomial coefficients are provided in the ancillary file in our repository submission[64].

The total loss of four-momentum at 5PM order, using equation (7), is given schematically by (see, for example, ref. 104):

$$P_{\mathrm{rad}}^{(5)\mu} = \frac{M^6 v^2}{b^5}\left([r_1(\gamma)\hat{b}^\mu + r_2(\gamma)(v_1^\mu - v_2^\mu)]\frac{m_1 - m_2}{M}\right.$$
$$\left. + [v_1^\mu + v_2^\mu][r_3(\gamma) + v r_4(\gamma)]\right). \tag{24}$$

Our 1SF/$\overline{\mathrm{ISF}}$ result fixes all coefficients except $r_4(\gamma)$. Similarly, we may derive the relative scattering angle[105,106] $\theta = \arccos(\mathbf{p}_{\mathrm{in}} \cdot \mathbf{p}_{\mathrm{out}}/|\mathbf{p}_{\mathrm{in}}||\mathbf{p}_{\mathrm{out}}|)$. Here $\mathbf{p}_{\mathrm{in}} = \mathbf{p}_1 = -\mathbf{p}_2$ is the incoming momentum in the centre-of-mass frame and

$$\mathbf{p}_{\mathrm{out}} = \mathbf{p}_{\mathrm{in}} + \Delta\mathbf{p}_1 + \frac{E_1}{E}\mathbf{P}_{\mathrm{recoil}} + \mathcal{O}(G^6), \tag{25}$$

is the (relative) outgoing momentum. We expand the scattering angle in $G$:

$$\theta = \Gamma \sum_{n=1}^{5}\left(\frac{GM}{b}\right)^n \sum_{m=0}^{\lfloor\frac{n-1}{2}\rfloor} v^m \theta^{(n,m)}(\gamma)$$
$$+ \frac{1}{\Gamma}\sum_{n=4}^{5}\left(\frac{GM}{b}\right)^n v^{n-2}\theta^{(n,n-2)}(\gamma) + \mathcal{O}(G^6). \tag{26}$$

Both the angle and $P_{\mathrm{rad}}^{(5)\mu}$ expansion coefficients are separated into parts even and odd under $v \to -v$ and expanded on suitable basis functions. All of these results can be found in the observables.m ancillary file in our repository submission[64].

## Checks

Our result for the impulse has been checked internally by the cancellation of $1/\epsilon$ poles and obeying the mass condition $p_1^2 = (p_1 + \Delta p_1)^2$ — verifying the $\{c_\nu', \bar{c}_\nu'\}$ coefficients (equations (21a) and (21b)). We have also checked that the (PPP) 1SF contribution is related to its $\overline{\text{1SF}}$ counterpart by symmetry. Furthermore, we have performed several checks using the non-relativistic ($\nu \to 0$) limit. First, PN results for the relative scattering angle at 5PM and to first order in self-force are known to 5.5PN order[104,105,107]. These include conservative terms at integer PN orders, starting from 0PN order, which were already matched in ref. 53, plus dissipative terms appearing at 2.5PN, 3.5PN, 4PN, 4.5PN, 5PN and 5.5PN orders, which we reproduce. Furthermore, and in the same works[104,105,107], dissipative PN results for the radiation of energy and recoil of the two-body system were reported at a relative 3PN order to their leading order. These allow us to perform non-trivial checks on $r_1(\gamma)$, $r_2(\gamma)$ and $r_3(\gamma)$ of equation (24) to relative 3PN order.

As an example of our results, we print explicitly the series expansion in $\nu$ of the $G^5$ component of $E_{\text{rad}}$:

$$E_{\text{rad}}^{(5)} = \frac{M^6 \nu^2}{\Gamma b^5}[(1-\gamma)r_2(\gamma) + (1+\gamma)r_3(\gamma) + \mathcal{O}(\nu)]$$

$$= \frac{M^6 \nu^2 \pi}{5\Gamma b^5 \nu^3}\left[122 + \frac{3{,}583}{56}\nu^2 + \frac{297\pi^2}{4}\nu^3 - \frac{71{,}471}{504}\nu^4\right.$$

$$+ \left(\frac{9{,}216}{7} - \frac{24{,}993\pi^2}{224}\right)\nu^5$$

$$+ \left(\frac{2{,}904{,}562{,}807}{6{,}899{,}200} + \frac{99\pi^2}{2} - \frac{10{,}593}{70}\log\frac{\nu}{2}\right)\nu^6 \qquad (27)$$

$$+ \left(\frac{7{,}296}{7} - \frac{2{,}927\pi^2}{28}\right)\nu^7$$

$$+ \left(\frac{4{,}924{,}457{,}539}{29{,}429{,}400} + \frac{8{,}301\pi^2}{112} - \frac{491{,}013}{3{,}920}\log\frac{\nu}{2}\right)\nu^8$$

$$\left.+ \left(\frac{99{,}524{,}416}{40{,}425} - \frac{46{,}290{,}891\pi^2}{157{,}696}\right)\nu^9 + \mathcal{O}(\nu^{10}, \nu)\right].$$

The terms in the square brackets up to $\nu^6$ reproduce the known PN results of $E_{\text{rad}}$, with the remaining three lines providing further hitherto unknown terms. Naturally, this series expansion can be extended to any order in $\nu$ using our present results. Explicit results relevant for the PN checks of the relative scattering angle and recoil are given in the Supplementary Information.

## Data availability

The complete set of observables up to the 5PM-1SF order of our work, the scattering angle, radiated energy and impulse four-vector are available in the Zenodo open repository submission https://doi.org/10.5281/zenodo.14604438 (ref. 64) accompanying this article. They are provided in the form of Mathematica computational notebooks. Source data are provided with this paper.

## Code availability

The analytical results of our work used the commercial computer algebra system Wolfram Mathematica[108], as well as the freely available computer algebra system Form[99]. The publicly available programs for reconstructing rational functions through finite-field sampling, FireFly[71,72] and FiniteFlow[85], were used. Furthermore, programs designed for the analytical treatment of differential equations, including MultivariateApart[86], CANONICA[80] and INITIAL[81], were used as described in Methods. Crucially, the IBP software package KIRA 3.0 (refs. 69,70)

was used for the reduction to master integrals in its beta version that can be made accessible by J.U. on reasonable request. For the numerical checks of the boundary integrals, the public software pySecDec[94–99] and AMPred[100–103] was used. All of our code and observables are provided in Mathematica computational notebooks: the WQFT Feynman rules necessary to perform the 5PM computation, the explicit form of the boundary integrals, as well as the 20 transcendental $G_i$ functions necessary to derive the canonical form of the differential equations. These computational notebooks are freely available in the Zenodo repository submission https://doi.org/10.5281/zenodo.14604438 (ref. 64). The set of IBP replacement rules to the master integrals (30 GB of code) generated using KIRA 3.0 can be made available on reasonable request to the corresponding author.

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

**Acknowledgements** We thank M. Beneke, W. Chen, C. Dlapa, R. Morales, R. Porto, N. Syrrakos, L. Tancredi, D. van Straten and M. Wilhelm for insightful discussions and J. Kleinmond for providing the waveform visualization of Fig. 1. This work was financed by the Deutsche Forschungsgemeinschaft (DFG, German Research Foundation) Projektnummer 417533893/ GRK2575 'Rethinking Quantum Field Theory' (G.U.J., G.M., B.S., J.P., J.U.) and 508889767/ FOR5372/1 'Modern Foundations of Scattering Amplitudes' (A.K., J.P.) and in part by the Excellence Cluster ORIGINS (C.N.) under Germany's Excellence Strategy – EXC-2094-390783311; the UK Royal Society under grant URF\R1\231578 'Gravitational Waves from Worldline Quantum Field Theory' (G.M.); the European Union through the European Research Council under grant ERC Advanced Grant 101097219 (GraWFTy) (M.D., J.P.); and ERC Starting Grant 949279 (HighPHun) (C.N.). The views and opinions expressed are, however, those of the authors only and do not necessarily reflect those of the European Union or the European Research Council Executive Agency. Neither the European Union nor the granting authority can be held responsible for them. This research was supported by the Munich Institute for Astro-, Particle and BioPhysics (MIAPbP) which is financed by the DFG under Germany's Excellence Strategy – EXC-2094-390783311 (G.U.J., G.M., J.P., B.S.). The authors gratefully acknowledge the computing time made available to them on the high-performance computer Lise at the NHR Center Zuse-Institut Berlin (ZIB). This centre is jointly supported by the Federal Ministry of Education and Research and the state governments participating in the National High-Performance Computing (NHR) joint funding programme (http://www.nhr-verein.de/en/our-partners).

**Author contributions** As is customary in high-energy physics, the authors and their contributions are listed in alphabetical order. M.D., G.U.J., G.M., J.P. and B.S. performed the integrand generation and planarization; M.D., B.S. and J.U. conducted the integration-by-parts reduction; C.N. and B.S. canonicalized the differential equations; A.K., C.N. and B.S. identified the underlying Calabi–Yau structures; M.D., G.U.J., G.M., J.P., B.S. and J.U. contributed to the boundary integrals and boundary matching; G.U.J., G.M. and B.S. performed the checks with the existing post-Newtonian results; M.D., G.U.J., C.N. and B.S. simplified the final results and implemented them numerically; G.M. and J.P. secured the funding and provided leadership of the project. All authors contributed to the writing of the manuscript.

**Funding** Open access funding provided by Humboldt-Universität zu Berlin.

**Competing interests** The authors declare no competing interests.

**Additional information**
**Correspondence and requests for materials** should be addressed to Jan Plefka.

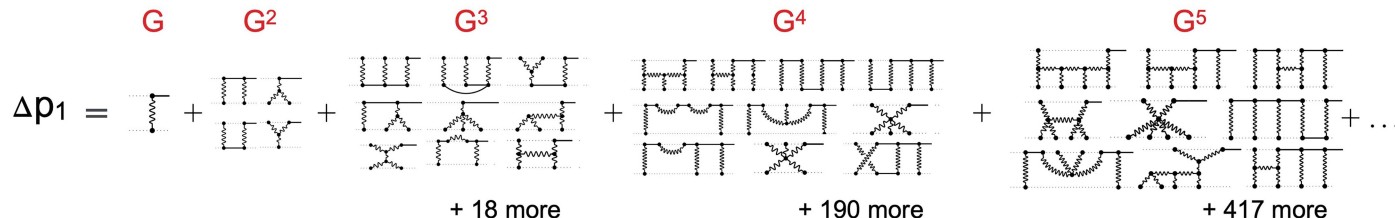

**Extended Data Fig. 1 | Expansion of the impulse $\Delta p_1$ in terms of Feynman diagrams.** Organized order by order in Newton's constant $G$. The dotted lines represent the worldlines of the two black holes, exchanging gravitons (wavy lines) and propagating deflection modes (solid lines).

$$\xrightarrow{\ell^\mu} = \frac{1}{\ell \cdot v_j + i0^+}$$

$$\xrightarrow{\ell^\mu}\!\!\!\!\text{wwww} = \frac{1}{(\ell^0 + i0^+)^2 - \boldsymbol{\ell}^2}$$

$$\cdots\cdots\overset{\ell^\mu}{\cdots}\cdots = \delta(\ell \cdot v_j)$$

$\sim G^{1/2}$

$\sim G$

$\sim G^{3/2}$

$\sim mG^{1/2}$

$\sim G^2$

$$= \int \mathrm{d}^D\ell \, \frac{\delta(\ell \cdot v_2)}{\ell^2 (q-\ell)^2 \ell \cdot v_1}$$

**Extended Data Fig. 2 | Feynman rules.** Retarded graviton, retarded worldline and cut worldline propagators, the relevant worldline and bulk vertices and a sample $G^2$ Feynman diagram. Here $\ell$ is the loop momentum and $q$ the momentum transfer.