## [Peer Review File · Nature]

Emergence of Calabi-Yau manifolds in high-precision black hole scattering

Corresponding Author: Professor Jan Plefka

This file contains all reviewer reports in order by version, followed by all author rebuttals in order by version. Additionally Reviewer #2 version 1 comments are attached at the end of the file.

Version 0:

Reviewer comments:

Referee #1

(Remarks to the Author)

Beyond its intrinsic academic significance in physics, the gravitational two-body problem occupies a foundational position in physics and astrophysics. Its importance has become increasingly pronounced with the rapid advancements in gravitational wave astronomy, spurred by the groundbreaking detections of gravitational waves from binary black hole and neutron star systems. As noted in the manuscript, while numerical relativity simulations are effective certain scenarios, they are computationally expensive and often inefficient for studying gravitational two-body inspirals or scattering in the weak-field regime. This limitation underscores critical need for analytic approximation methods, which are indispensable for both theoretical explorations and experimental observations. In practice, the synergy between numerical and analytic methods is crucial for generating the vast array of waveform templates—key tools in advancing gravitational wave astronomy.

Within the framework of perturbative methods, calculations are systematically organized as series expansions in Newton constant (or other small parameters), known as the post-Minkowskian (PM) expansion. In this expansion, the expression for the two-body dynamics is naturally a polynomial in the two masses (or their mass ratio), which may be understood as charges of gravitational forces. The PM expansion is structured such that the first and second PM orders are fully captured by the leading term in the mass ratio (0SF), the third and fourth PM orders incorporate both leading and sub-leading mass ratio contributions (1SF), while the fifth PM order includes terms up to third order in the mass ratio (2SF). The authors reported a new computation of the gravitational impulse—the momentum change experienced by each body during a scattering event—alongside the associated scattering angle and radiated energy. Earlier this year, many of the authors completed a computation for the conservative dynamics at the fifth PM order to 1SF; they have published the evaluation of the needed Feynman integrals involving Calabi-Yau manifolds [Phys. Rev. D 109 (2024) 124046] and the analytic result for observables [Phys. Rev. Lett. 132 (2024) 241402]. In this manuscript, they further extend their analysis to the dissipative contribution to the gravitational impulse as well as resulting scattering angle and radiated energy, thus completing the knowledge of gravitational two-body dynamics at the fifth PM order to 1SF precision. As such, this work represents a landmark achievement in the domain of analytic relativity.

In my opinion, as mentioned above, the new result itself (which the authors describe as "highest-precision") holds significant scientific value in General Relativity. However, the authors place considerable emphasis on the appearance of Calabi-Yau manifolds in their calculations, highlighting their claim that these mathematical objects "appear for the first time in nature". While this aspect is fascinating indeed, it is essential to prioritize the core scientific contributions and their broader impact. Nature values "originality, importance, interdisciplinary interest, timeliness, accessibility, elegance and surprising conclusions" in a manuscript. Moreover, while it may indeed be rare for analytic results in particle physics to explicitly involve Calabi-Yau manifolds, it is well-known—and not surprising—that many Feynman integrals can be ultimately connected to Calabi-Yau geometry.

The methodology adopted in this work, combining Effective Field Theory approach and modern Feynman loop integral techniques, is well-established. The same methodology has been instrumental in deriving gravitational two-body dynamics at the third and fourth PM orders, as well as in many state-of-the-art Post-Newtonian calculations. While the authors overly emphasize their so-called WQFT, it is essentially a variant of the worldline Effective Field Theory approach for gravitationally extended objects, originally introduced by Goldberger and Rothstein [Phys. Rev. D 73 (2006) 104029]. Notably, this manuscript's authors even didn't reference this foundational paper in their manuscript. The evaluation of Feynman loop integrals leverages standard techniques such as integration-by-parts reduction, differential equations, and the method of

regions—all of which are massively studied and well-established in particle physics. Therefore, the methodology adopted in this study is robust, and the reported results are reliable and credible. Most impressively, the authors succeeded in solving differential equations of Calabi-Yau threefolds. They clearly outlined the procedure of transforming these differential equations into epsilon-forms and introduced new functions. They also kindly provided electronic files for all key results in a repository, ensuring transparency and accessibility for the community.

Overall, the manuscript is well-written, with a clear and accessible abstract. However, due to the specialist nature of perturbative quantum field theory, some parts of the presentation may be overly technical, potentially limiting accessibility for a significant portion of Nature's readership.

(Remarks on code availability)

All key results presented in the manuscript are stored in the repository, accompanied by a README file that clearly outlines the structure, purpose, and content of the various files and expressions. I have carefully reviewed the repository and checked its electronic data against the manuscript. Based on my assessment, all provided files and results appear to be consistent with the findings and claims reported in the manuscript.

Referee #2

(Remarks to the Author)

Please see file at the end of the document.

Referee #3

(Remarks to the Author)

Review of High-precision black hole scattering with Calabi-Yau manifolds
by M. Driesse et al

This article reports on a landmark result that is likely to stay, for many years to come, as one of the highlights of a currently very active area of research. The main new results concern the motion, and the gravitational radiation, of a binary system of two point masses (interacting via Einsteinian gravity), here obtained at an unprecedented analytical accuracy, by importing integration techniques from collider physics, together with sophisticated results of algebraic geometry. Though these new analytical results, as they are, cannot be immediately used to facilitate the detection of new gravitational-wave sources (as the authors recognize), they bring a new level of analytical understanding of the gravitational interaction of, and gravitational radiation from, binary black holes, which is likely to have a significant, and lasting, impact on the future development of more accurate gravitational-wave detection templates.

In addition, though the results reported here are based on quite complex physical and mathematical concepts and results, the authors made quite an effort to present the essence of their work, and its conceptual beauty, in a streamlined fashion, potentially accessible to a large audience of physicists.

I will be happy to warmly recommend publication of this work in Nature, after the authors consider my comments below, aimed either at clarifying the content of the paper, or at improving its readability for non-expert readers.

1. The abstract writes that « elliptic integrals made their first appearance in the Newtonian two-body problem ».

This not an accurate statement.

Elliptic integrals appeared when trying to compute the length of an ellipse.

[This computation never enters the Newtonian 2-body problem.]

In classical mechanics, they appear when computing the period of a spherical (or circular) pendulum (or when computing the scattering angle of a probe mass around a Schwarzschild black hole).

2. page 1 the text writes: « The first non-trivial order for BH scattering (G^2) was found in 1979²⁵ » where Ref. 25 refers to a 1979 paper of Westpfahl+Goller.

Actually, the latter paper does not compute the scattering of two BHs, but rather gives the 2PM equations of motion.

The first computation of the G^2 scattering was published by {Westpfahl:1985tsl}.

3. I am not happy with Fig. 2 and the corresponding caption and text.

One of the conceptual beauties of the present work is the appearance of CY periods. I encourage the authors to explain in a more pedagogical manner the precise meaning, and progression in complexity, between elliptic integrals, K3 periods and CY3 periods. E.g. the authors could start recalling the formulas for the usual (Legendre-form) complete elliptic integrals

$\int_a^b dx/y$ based on a cubic curve

$y^2=x(x-1)(x-z)$

with an integral taken either from 0 to 1 or 1 to z (or whatever).

Then Fig. 1 could feature the Legendre cubic curve (as a real planar curve) and the text explain that the integral of the (holomorphic) 1-form $\omega_1=dx/y$ on independent cycles (when viewing the curve, a la Riemann as a real 2-surface) gets generalized to integrals of (holomorphic) 2-forms over K3 surface, and integrals of (holomorphic) 3-forms of CY3. The current Fig 1 seems to mix real (n-dimensional) sections of complex varieties (of complex dimension n) with (for n=1) 2n-dimensional real representations. I find this more confusing than clarifying.

In addition, in preparation for the K3 and CY3 Picard-Fuchs ODEs, the authors could recall that complete elliptic integrals satisfy a well-known first-order system (first derived by Legendre) leading to a second-order ODE for each elliptic period.

The motivation for the above suggestions is that I think that a general physicist has some knowledge of elliptic integrals (and their link to an holomorphic 1-form dx/y) so that such pedagogical reminders will help many readers to better grasp the meaning of the K3 and CY3 periods entering your results.

4. p 4 It would be good to write « In the velocity-odd » sector of integrals (instead of « in the odd sector »)

5. p 4 near Eq (5). As the appearance of a CY3 is conceptually novel in GW physics, I would encourage the authors to describe in more details the algebraic equations defining their CY3. Can one write some explicit equation similar to a quintic (containing x as unique parameter in its coefficients) in CP^4 , together with an explicit form of the (holomorphic)3-form ω_3 ?

Then, after writing the Picard-Fuchs equation the authors might mention that one solution of this equation is a (rather simple) generalized hypergeometric function ${}_pF_q[a, \dots, b, \dots, x]$.

I asked Mathematical DSolve to solve eq (5) and

immediately got concrete solutions. Notably

$(-1)^{1/4} x C[1] \text{HypergeometricPFQ}[\{1/2, 1/2, 1/2, 1/2\}, \{1, 1, 1\}, x^4]$.

Indeed, I think that it would be good

for the reader to have in mind some explicit expression

(in ${}_pF_q$ form) for at least one of the periods, so that (s)he has a better

feeling for the complexity of later using such a function inside an iterated integral

6. p 4 below eq 6 « rescaled energy » \longrightarrow « mass-rescaled energy »

7. p 5 Fig 6 and its caption:

I do not understand how Fig 6 was made. First, the caption says that it uses an NR simulation from Ref 63 with $v=0.5$. But, I do not see any such simulation in Ref. 63.

The closest relative velocity would be for

the case that $E_{in} = 1.04033$ in Table I, corresponding to $v=0.5125$ (which is significantly different from 0.5). In addition, the corresponding rescaled impact parameters b/GM are 16,14,12,10,9.6,9.5,9.4,9,8.8.

The corresponding scattering angles seem to be those plotted in Fig. 6,

but I do not understand why the impact parameters b/GM used in Fig 6

are not those listed above. They seem to differ by some factor??

Am I saying something wrong? Anyway, the reader needs more explanations to understand how Fig. 6 was made.

8. p 5 Fig. 7 and its caption.

I do not understand what is plotted in Fig. 7. What is this rescaling by

« the leading non relativistic value up to the G^5 » ???

I would like to make a suggestion to clarify the physical content of the new results obtained by the authors concerning the G^5 radiated energy-momentum.

Indeed, as the authors mention in their Eq. 24,

the full content of the G^5 radiated energy-momentum is described by four functions of γ , which they denote $r_1(\gamma)$, $r_2(\gamma)$, $r_3(\gamma)$ and $r_4(\gamma)$ (the latter being at the 2SF level).

[Let me mention in passing that this was explicitly pointed out

in Ref. 95 (=BDG2023), see Eq 11.8 of [95], involving the

functions

$p_{[2/2]^{1+2,G^5}}(\gamma, \nu)$ which is linear in ν ,

$p_{[1/2]^{1-2,G^5}}(\gamma)$, and $p_{[1/2]^{b12,G^5}}(\gamma)$.]

I would then suggest that the authors replace (or complete?) Fig 7

by a Figure displaying the three building blocks they determined, namely $r_1(\gamma)$, $r_2(\gamma)$, and $r_3(\gamma)$ as functions of γ (or ν).

[Note also that Eq. 11.28 of [95] has displayed the known PN knowledge of these functions.]

Independently of the issue of displaying the results of the authors in some figure, I expected the authors to discuss the high-energy behavior of their Erad results. Does it worsen the issue found at G^3 and G^4 that the perturbative Erad exceeds (at high γ 's) the incoming energy even when considering a small, fixed scattering angle?

9. p 5 In eq (8) should there not be a factor $1/2$??

10. p 9 I would like to remind the authors that the definition of the relative scattering angle (with Eq. 25) was introduced in Ref 94 (= BDG2021)

11. p 9 second column: The authors mention that they have checked that their G^5 scattering results agree with PN results up to the 5PN order. Let me mention that more PN terms in the (radiation-reacted) scattering have been derived. In particular, results in the PN literature yield a *complete* answer for the radiation-reacted scattering *at the 5.5PN level* (i.e. one power of v beyond the 5PN level). Indeed, Ref. Bibitem{Bini:2020hmy} D.~Bini, T.~Damour and A.~Geralico, %``Sixth post-Newtonian nonlocal-in-time dynamics of binary systems," Phys. Rev. D \textbf{102}, no.8, 084047 (2020) has derived the 5.5PN *conservative* contribution to the relative scattering angle. Interestingly, this 5.5PN contribution is due to subtle tail-of-tail (or tail²) nonlocal interactions; see Eq. 5.10 there. This conservative contribution to scattering can then be completed by corresponding radiation-reaction contributions which have been derived in [94] and [95] (BDG2021 and BDG2023). See notably Table IX of [94] and Table II of [95], as well as Sections 12 I and 12 J of [95] which also completed the linear-response scattering formula for the (relative) scattering angle, by deriving the fully nonlinear radiation-reaction contributions to the G^5 impulse modulo a 2SF contribution in the longitudinal component of the impulse.

In particular, I have checked that by combining the above-cited PN results one exactly reproduces the value of the 1SF relative scattering angle, i.e. modulo a rescaling, the quantity $\theta^{(5,1)}$ (Eq S.3a of the Supplementary information), up to the 5.5PN level (i.e. up to the v^1 term).

[By contrast, the 6PN-level results derived in the above references do not yield (even at 1SF) a complete radiation-reacted result to be compared with the 1SF G^5 results presented by the authors.]

I encourage the authors to redo this 5.5PN check (which brings a further deep confirmation of their results, in view of the many tail, and tail², effects included in the needed PN results), or at least to mention that complete 5.5PN results exist in the literature.

12. SUPPLEMENTARY INFORMATION

Eqs S.3, see above concerning the 5.5PN check of the scattering

Thibault Damour

(Remarks on code availability)

I did not spend time on downloading the full code. I saw parts of it. Everything looked clear to me. And it is very useful to make such a code available to other authors.

Referee #4

(Remarks to the Author)

The current paper uses effective field theory to describe the scattering process of two gravitationally interacted particles. Along the EFT framework, the authors expand the solution as a series respect to the gravitational constant G . With 'high-precision', the authors means they did calculation till G^5 . It is interesting and important to get such high order result. In addition, the authors find that the results show an interesting special function which is related to CY manifold. The appearance of such a special function give us insight of the character of gravity. These results are interesting and important enough for publication in Nature.

As a Nature paper, I have several comments and suggestions in the following.

1. If the two particles mean matter such as neutron star, there is direct meaning of mass (m_1, m_2). But if pure black holes are involved, the masses are reduced concepts ($R/2G$) where R is the characteristic size respect to the spacetime curvature. Consequently the action can be factorized as $1/G$ and another G independent expression. This leads us to Einstein equation without G . In this situation, the G series do not make sense. In fact the expansion is essentially respect to R/d where d is the separation of the two objects. Then conceptually I suggest the authors explain the difference between EFT and usual PN approximations.

2. For the faced problem, it is a standard classical problem. In this sence the action (1) is exact. There is no effective meaning at all. I just guess the authors call the theoretical framework EFT just because they treat it as a quantum field

theory. And consequently the Feynman diagram technique is applicable. Then I have a question. What kind of rules guarantee the quantum field treatment can result in the solution of the original standard classical problem? For comparison, we have EOBNR models for CBC systems. EOBNR theory also uses quantum field theory technique. But we have to notice that EOB just gives better convergent result than PN theory. Theoretically we are not sure EOB results corresponds to the real solution of Einstein equation. Only after the calibration by numerical relativity, we are sure the EOB results is trustable. Regarding the EFT involved in the current paper, I suggest the authors give some comments on this point.

3. There is one more question about gauge problem in general relativity. Along with the EFT, the authors work in some coordinate where the spacetime metric takes form $g_{\mu\nu}$. I imagine that initially ($t \rightarrow -\infty$) $g_{\mu\nu}$ is Minkowsky. My concern is final ($t \rightarrow +\infty$) state of $g_{\mu\nu}$. If final $g_{\mu\nu}$ is Minkowsky, the meaning of angles is straightforward and the presentation of the current paper is without problem. If it is not Minkowsky, the metric should affect the geometric meaning. Here with 'Minkowsky' I mean the metric form of Minkowsky under the inertial coordinates, $\eta_{\mu\nu}$. In general, the gravitational memory may lead the final metric do not equal to $\eta_{\mu\nu}$ although we are sure the final spacetime is Minkowsky. Since the accuracy of the current paper is extremely high, how about this effect? Or to say the memory effect is higher than G^5 order?

4. As the authors said, the CY n-folds "generalize the elliptic integrals and are known to arise in perturbative QFT in multiloop Feynman integrals". Does this mean the CY n-folds appear for any perturbative QFT instead of specifically for the EFT of GR?

5. The authors emphasized that "Until now, no physical observables in elementary particle physics or gravity have been reported where CY3 periods appear." Then does the result shown in the current paper mean we can use binary black hole observations to relate CY3?

(Remarks on code availability)

The code includes several mathematica files. And a readme file is also included. The information is complete. The description is clear. These mathematica files are helpful to our readers who are interested in extending the related research.

Referee #5

None

Version 1:

Reviewer comments:

Referee #1

(Remarks to the Author)

I have carefully read the revised manuscript as well as the authors' responses. As a referee, I also gained a lot from this process, and would like to express my gratitude to the authors and the other anonymous referees. Below are my further comments:

1. I fully agree with and appreciate the authors' strategy of presenting the material in a "technically graded" manner. However, to maximize the accessibility of the paper, it is essential to provide clear and intuitive explanations of key mathematical structures—such as elliptic curves, K3 surfaces, and Calabi-Yau manifolds—particularly in the introductory part of the paper.

2. Following my earlier suggestion, the authors have cited Goldberger and Rothstein's seminal work on the worldline EFT approach in their revised version.

3. The authors observed that CY integrals "have so far not appeared in a final observable," but their explanation is somewhat oversimplified. The claim that "the most complicated structures of the related Feynman integrals drop out" in amplitudes is generally valid for relatively simple theories (and processes). In more complex theories, such as the electroweak theory, the presence of multiple scales—such as the masses of Higgs and W/Z bosons—can lead to results that involve elliptic/CY integrals and even more complicated functions. In such cases, observables are often best expressed numerically rather than in terms of huge analytic expressions involving complicated functions. While I am not happy with "...Calabi-Yau manifolds appear for the first time in nature", I do acknowledge that classical two-body scattering, due to the simplicity of its final results, provides an excellent testing ground for exploring the interplay between physics and Calabi-Yau geometry.

(Remarks on code availability)

The authors did not change the electronic attachment in this reversed version.

Referee #2

(Remarks to the Author)

Please see file at the end of the document.

Referee #3

(Remarks to the Author)

All my suggestions were adequately addressed (sometimes with an understandable constraint coming from page limits) in this revised version. I have also looked at the modifications aimed at addressing the comments of the other referees. I think these modifications satisfactorily addressed the comments of the other referees, and that the paper is now improved, and will be understandable, and of interest, to a large readership. I warmly recommend publication of this landmark paper as is.

Thibault Damour

(Remarks on code availability)

I had a quick look at the code, which looked very clear to me, without trying to run it.

Referee #4

(Remarks to the Author)

Thanks the authors for their careful checking of my concerns. After reading the revised paper I have one more question. Since the period of CY manifolds appeared in GR scattering is the most important finding of the current paper, I suggest the authors to explain these quantities in a simpler way.

CY manifolds can be definitely scaled. So I guess the specific periods are not fixed numbers. Instead some ratio may be a geometric quantity. For an example, torius admits π which corresponds to the period of toruses. I guess K3 and CY3 may admit some similar quantities related to the periods mentioned by the authors. But maybe I am wrong. Anyhow, if the authors can explain the periods in this kind of simpler way, I believe our readers can benefit more from the paper.

In conclusion, I am happy to recommend this paper for publication in Nature because it's interesting and important enough.

Referee #5

(Remarks to the Author)

None

Referee #1 (Remarks to the Author):

Beyond its intrinsic academic significance in physics, the gravitational two-body problem occupies a foundational position in physics and astrophysics. Its importance has become increasingly pronounced with the rapid advancements in gravitational wave astronomy, spurred by the groundbreaking detections of gravitational waves from binary black hole and neutron star systems. As noted in the manuscript, while numerical relativity simulations are effective certain scenarios, they are computationally expensive and often inefficient for studying gravitational two-body inspirals or scattering in the weak-field regime. This limitation underscores critical need for analytic approximation methods, which are indispensable for both theoretical explorations and experimental observations. In practice, the synergy between numerical and analytic methods is crucial for generating the vast array of waveform templates—key tools in advancing gravitational wave astronomy.

Within the framework of perturbative methods, calculations are systematically organized as series expansions in Newton constant (or other small parameters), known as the post-Minkowskian (PM) expansion. In this expansion, the expression for the two-body dynamics is naturally a polynomial in the two masses (or their mass ratio), which may be understood as charges of gravitational forces. The PM expansion is structured such that the first and second PM orders are fully captured by the leading term in the mass ratio (0SF), the third and fourth PM orders incorporate both leading and sub-leading mass ratio contributions (1SF), while the fifth PM order includes terms up to third order in the mass ratio (2SF). The authors reported a new computation of the gravitational impulse—the momentum change experienced by each body during a scattering event—alongside the associated scattering angle and radiated energy. Earlier this year, many of the authors completed a computation for the conservative dynamics at the fifth PM order to 1SF; they have published the evaluation of the needed Feynman integrals involving Calabi-Yau manifolds [Phys. Rev. D 109 (2024) 124046] and the analytic result for observables [Phys. Rev. Lett. 132 (2024) 241402]. In this manuscript, they further extend their analysis to the dissipative contribution to the gravitational impulse as well as resulting scattering angle and radiated energy, thus completing the knowledge of gravitational two-body dynamics at the fifth PM order to 1SF precision. As such, this work represents a landmark achievement in the domain of analytic relativity.

We thank the referee for this judgement of our work.

In my opinion, as mentioned above, the new result itself (which the authors describe as "highest-precision") holds significant scientific value in General Relativity. However, the authors place considerable emphasis on the appearance of Calabi-Yau manifolds in their calculations, highlighting their claim that these mathematical objects "appear for the first time in nature". While this aspect is fascinating indeed, it is essential to prioritize the core scientific contributions and their broader impact. Nature values "originality, importance, interdisciplinary interest, timeliness, accessibility, elegance and surprising conclusions" in a manuscript. Moreover, while it may indeed be rare for analytic results in particle physics to explicitly involve Calabi-Yau manifolds, it is well-known—and not surprising—that many Feynman integrals can be ultimately connected to Calabi-Yau geometry.

We have clarified the fact that the appearance of CYs is not fully unexpected yet reflecting the comments above through a number of changes in the two paragraphs on page 2 where CY periods are introduced. All changes to the original manuscript are typeset in red.

Indeed, while the observation of a CY in individual Feynman integrals has been known for 5y now, they have so far not appeared in a final *observable*. The reason being that in particle physics scattering amplitudes are generically significantly simpler than individual Feynman integrals, the *raison d'être* of the very successful generalized unitarity program. There are many examples known

where, in the final amplitude, the most complicated structures of the related Feynman integrals drop out. This observation has also been made in recent gravity computations, e.g. our conservative 5PM-1SF result or the analogue one in N=8 gravity of Bern et. al. where the elliptic integrals related to a K3 cancel in the scattering angle. This is why we do emphasize in the paper that our computation is the first one in perturbative QFT where a full-fledged CY3 period appears in the final physical result. We believe that this highlights the growing importance of CYs in physics, for the first time observed in our classical gravity computation and soon expected to emerge in precision calculations in particle physics.

We have also now motivated more why CYs appear in physics as they lie in a certain sense at the border between simplicity and complexity – despite the fact that a full understanding of this fact remains elusive.

The methodology adopted in this work, combining Effective Field Theory approach and modern Feynman loop integral techniques, is well-established. The same methodology has been instrumental in deriving gravitational two-body dynamics at the third and fourth PM orders, as well as in many state-of-the-art Post-Newtonian calculations. While the authors overly emphasize their so-called WQFT, it is essentially a variant of the worldline Effective Field Theory approach for gravitationally extended objects, originally introduced by Goldberger and Rothstein [Phys. Rev. D 73 (2006) 104029]. Notably, this manuscript's authors even didn't reference this foundational paper in their manuscript. The evaluation of Feynman loop integrals leverages standard techniques such as integration-by-parts reduction, differential equations, and the method of regions—all of which are massively studied and well-established in particle physics. Therefore, the methodology adopted in this study is robust, and the reported results are reliable and credible. Most impressively, the authors succeeded in solving differential equations of Calabi-Yau threefolds. They clearly outlined the procedure of transforming these differential equations into epsilon-forms and introduced new functions. They also kindly provided electronic files for all key results in a repository, ensuring transparency and accessibility for the community.

Again, we thank the referee for this kind assessment. We apologize for the omission of the seminal work of Goldberger & Rothstein and agree with the referee on the fact that our formalism derives from WEFT. We now cite Goldberger & Rothstein [20] as the seminal paper introducing the EFT & diagrammatic approach to the two-body problem in the introductory paragraphs on page 1 when introducing the QFT approach. Moreover, we have included a clear reference to WEFT as being the intellectual basis of our approach when we introduce WQFT on page 3, again citing this work. (all changes to the initial manuscript are typeset in red)

Overall, the manuscript is well-written, with a clear and accessible abstract. However, due to the specialist nature of perturbative quantum field theory, some parts of the presentation may be overly technical, potentially limiting accessibility for a significant portion of Nature's readership.

Our presentation is technically graded: The main part should be accessible to a wider audience and technicality increases in the Methods and Supplementary Parts. We believe that together with the figures and graphs the main message is communicated in the main part.

Referee #2 & #5 (Remarks to the Author):

This work presents the first complete calculation of scattering observables for binary compact objects, modeled as point particles, in the post-Minkowskian (PM) expansion up to the fifth order in the gravitational constant G and the second order in the symmetric mass ratio ν . The authors compute the change in four-momentum, the scattering angle, and the energy radiated by gravitational waves (GWs) up to $O(G^5)$ and $O(\nu^2)$, achieving agreement with lower-order PM results, post-Newtonian (PN) expansions, and numerical relativity simulations. While focused on unbounded orbits, their results can potentially provide valuable input to other endeavors, such as the effective-one-body formalism, for modeling the GWs emitted by binary inspirals (with bounded orbits), which are the main targets of LVK detectors. Novel mathematical features, such as high-fold Calabi-Yau (CY) periods, also emerge in their observables. These new geometrical quantities may reveal interesting connections between classical scattering problems and geometries underlying quantum gravity theories, such as string theory.

This work is technically sound and has practical importance for developing efficient and high-precision waveforms required by next-generation GW detectors. This work focuses on employing cutting-edge techniques, such as integration-by-parts reduction, to tackle the computational challenges of evaluating a vast number of Feynman integrals.

We thank the referees for this positive judgement of our work.

These techniques, while impressive, also raise questions of whether this work is of enough interest to a broader community. Many of these techniques seem to have mature packages or algorithms developed for more general problems in differential equations. While the authors have made novel improvements, such as methods for decoupling differential equation sectors with different CY periods, the practical utility of these advances outside gravitational scattering remains unclear.

In fact, our methodology for decoupling the CY3 and K3 sectors and integrating the complete problem is of relevance beyond the classical gravitational scattering problem. CY periods have appeared in Feynman integrals relevant to elementary particle physics, e.g., the electron and photon self-energy in QED at three-loops (c.p. the recent arxiv.org 2408.05154 and 2411.19042), and the methodology used here was applied. Therefore, the methods developed in this work will be beneficial for future high-precision studies in collider physics. To clarify this, we have added a sentence on page 2 after introducing CYs and in the outlook of the main text making this point. (all changes are typeset in red)

Furthermore, all these technical details also hide what new physics the authors have learned or could potentially learn by conducting these high-order PM calculations besides obtaining more precise waveforms. For example, this work emphasizes the novel appearance of the CY3 periods in their observables. However, whether different CY periods could be related to specific observables in GW detections is unclear. If they could, how would they concretely further our understanding of quantum gravity due to their connections to string theory?

We thank the referees for raising this point. The CY3 periods enter the radiated energy and three-momentum recoil at the 5PM-1SF order. In these observables, they contribute to the tail-of-tail effect, i.e., the backscattering of radiative gravitons off the gravitational potential in the space-time bulk. In that sense one may attribute their appearance to a physical effect. We have included a passage in the final discussion on the last page of the main text raising

this point. Furthermore, we already know that at the 5PM-2SF order - a future objective - a different CY3 geometry appears in the integration problem - yet we do not know whether it will contribute to the final observables. We have added a statement in the final summarizing paragraph to make this point clearer.

It would be very interesting to have an understanding which CY geometries appear at which order in perturbation theory. Yet, this is beyond the scope of this work. An interesting observation is that the same differential operator, $L_2 := (x \frac{d}{dx})^2 - x^2 (x \frac{d}{dx} + 1/2)^2$, appears in the K3 (as symmetric square) and CY3 (as Hadamard product) at 4PM-1SF and 5PM-1SF. It remains to be seen, whether L_2 continues to be the "building block" at higher PM 1SF orders and in higher CY geometries. This fact was already commented on in the previous work of some of us [Phys. Rev. D 109 (2024) 124046] and we now mention this intriguing fact in footnote 1 on page 4.

Whether the appearance of CY geometries in classical observables that we report on in our work has any connection to quantum gravity or string theory is, however, mere speculation. There is a priori no reason for such a connection, as the appearance of CY3s in string theory arises from the compactified six dimensions of space-time. The 4d classical gravity two-body problem studied here has no connection to such an underlying quantum structure. Put differently, irrespective of whether the underlying theory of quantum gravity is string theory or something else, our classical physics results are unchanged.

For these reasons, this work may not be suitable for publication in Nature. Despite these concerns, the results in this work are significant, and the manuscript is well-written so that it could get published in, for example, other journals with a more specific focus.

We believe that our results are significant enough to appear in Nature. They represent a "landmark achievement in the domain of analytic relativity" to quote referee #1 and we believe that our presentation will find interest in a wider audience. The fact that a CY3 period appears in a physical result of a black hole scattering is interesting in itself – even if the technical details of the computation might not cater to everybody. This is what we convey in the main text.

Below we include more specific, minor comments/suggestions on the manuscript:

1. In the second and third paragraphs of page 1, the authors briefly review various methods for waveform modeling and argue analytical methods can help reduce the high computational expense of numerical relativity. While the progress in PM formalism has been discussed in detail, the authors could also add more discussion on the current stages of other analytical or semi-analytical approaches, such as black hole perturbation theory, gravitational self-force, and PN formalism. This will help the readers understand where this work sits in the broader efforts of waveform modeling.

We have now mentioned the PN and gravitational self-force formalisms in the introductory section of the main text with citations to a selection of relevant review papers. We hope that the referee deems this sufficient (all changes to the manuscript are typeset in red). We feel that a more detailed discussion of the present state-of-the-art in these two approaches would require more technicalities at an early stage, thereby making the manuscript less accessible. Also we are limited by the space available. Similarly, the black hole perturbation theory is relevant for the ringdown phase of the binary, yet our focus is on the scattering scenario. This is why we chose not to discuss it here.

2. At the end of page 1, the authors mentioned that at least the $O(G^5)$ precision will be needed for GW detectors. Could the authors also comment

on what PM order will be enough for next-generation detectors? Suppose $O(G^5)$ is not even sufficient. How hard will it be to extend this calculation to higher PM orders, especially compared to other efforts to reduce the computational cost of waveform modeling, such as developing more efficient numerical relativity algorithms or surrogate models?

The statement that “at least the fifth order (G^5) precision” will be needed in the manuscript does refer to the third or next-generation of GW detectors to go online in the 2030s.

The next steps in complexity are either the 5PM-2SF computation, now giving rise to non-planar graphs, or the 6PM-1SF order. We believe that the integral family (10a) generalized to n -loops will be able to cover all $(n+1)$ PM-1SF integrals, i.e. it is the complete 1SF integral family (this was conjectured by many of us in [Phys. Rev. Lett. 132 (2024) 241402]). Yet, the IBP problem at 6PM-1SF and higher remains the bottleneck and it is hard to estimate the computational costs (understood as core hours CPU time) thereof without having done pilot computations.

As for the 5PM-2SF problem, we have already quantified the computational costs to be 19Mcore-h as we are presently pursuing this goal and have applied for HPC time. How does this compare to numerical relativity? Here a single high-resolution run costs around (10-50)kcore-h (Oliver Long private communication). Yet, as we need to populate a template databases of tens of millions of entries, even the analytical costs of 19Mcore-h would amount to a factor of 10,000 in favor of analytical results. Here of course, only one run is necessary to obtain a final analytical (yet complex) expression. Effective-one-body models built from our analytical one-time run data will be able to fill these for all desired configurations, whereas in numerical relativity, every single configuration needs to be rerun at the above costs. In addition, situations of large mass hierarchies - as are relevant for our 1SF result - are very costly in numerical relativity, as are long inspiral phases.

In summary, this demonstrated that at least for the 5PM-2SF order, the analytical approach is still computationally advantageous over a purely numerical based one - in particular for the inspiral phase.

3. On page 2, the authors discuss that the K3 periods have been encountered at $O(G^4)$, and the CY3 period appear when the calculations are extended to $O(G^5)$, as in this work. In this case, shall one expect higher-fold CY periods to appear for calculations at a higher PM order? For $O(G^n)$ corrections, shall one always expect $(n-2)$ -fold CY period to appear? Or could the structure differ?

This is a very good question. Generically, one would expect such a structure, i.e. G^n can give rise to CY $(n-2)$ folds based on transcendentality arguments, we have added a statement on page 2 on this. In addition, it is known that periods on higher genus Riemann surfaces also appear, but there are indications that they may be expressible via CY periods as well. Yet no proofs exist. Today, no other structures are known to appear in Feynman integrals.

4. As mentioned above, are these additional high-fold CY periods related to new physical features when improving the description of scattering encounters? If they are, some explanation of the correspondence of the first few CY periods to physical observables might help the readers have a clearer physical picture.

Please see our answer to the general point above (our second answer) where we explain the tail-of-tail effect being tied to the CY3 appearance. This has been reflected now in an addition to the main text in the final discussion. This structure could persist to higher orders, i.e. the tail-of-tail-of-tail..

effects could account for higher CY n-folds. We would at this point yet not speculate along these lines.

5. In Fig. 7, it seems that the additional radiated energy due to $O(G_5)$ corrections is comparable to the total energy radiated up to $O(G_3)$. Since PM formalism is a perturbative approach, should one expect the correction to the radiated energy to decrease generally at a higher PM order, so that the radial energy converges?

In reaction to the requests of referee #4 we have decided to delete the plot of Fig. 7. The plot used an intricate G-dependent normalization of the plotted G-orders for the radiated energy in order to display differences. This leads to ill interpretations.

6. Could the authors more clearly define the symbols l_1, \dots, l_3 above Eq. (13)? If they are defined in the earlier part of the paper, the authors may consider briefly reviewing the definition here.

Done, see revised manuscript.

7. In Eq. (13), the symbol θ is defined to be $x \frac{d}{dx}$, while it was also used as the scattering angle in other parts of this paper. Could the authors choose a different symbol for $x \frac{d}{dx}$ to avoid confusion?

We thank the referees for pointing this out. We introduced $\hat{\theta}$ as the operator representing $x \frac{d}{dx}$ above (13). The use of θ for the logarithmic derivative is standard in the mathematical community in this context, which is why we would like to maintain this symbol in the methods section.

8. Could the authors clarify the meaning of “lower $G_i(x)$ ” functions above Eq. (17)? Does it mean that, for example, the differential equation of $G_4(x)$ will depend on $G_1(x), \dots, G_3(x)$?

Done, see revised manuscript.

Referee #3 (Remarks to the Author)

This article reports on a landmark result that is likely to stay, for many years to come, as one of the highlights of a currently very active area of research.

The main new results concern the motion, and the gravitational radiation, of a binary system of two point masses (interacting via Einsteinian gravity), here obtained at an unprecedented analytical accuracy, by importing integration techniques from collider physics, together with sophisticated results of algebraic geometry. Though these new analytical results, as they are, cannot be immediately used to facilitate the detection of new gravitational-wave sources (as the authors recognize), they bring a new level of analytical understanding of the gravitational interaction of, and gravitational radiation from, binary black holes, which is likely to have a significant, and lasting, impact on the future development of more accurate gravitational-wave detection templates.

In addition, though the results reported here are based on quite complex physical and mathematical concepts and results, the authors made quite an effort to present the essence of their work, and its conceptual beauty, in a streamlined fashion, potentially accessible to a large audience of physicists.

I will be happy to warmly recommend publication of this work in *Nature*, after the authors consider my comments below, aimed either at clarifying the content of the paper, or at improving its readability for non-expert readers.

We thank the referee for this warm review of our work!

4.1) The abstract writes that elliptic integrals made their first appearance in the Newtonian two-body problem.

This not an accurate statement.

Elliptic integrals appeared when trying to compute the length of an ellipse.

[This computation never enters the Newtonian 2-body problem.]

In classical mechanics, they appear when computing the period of a spherical (or circular) pendulum (or when computing the scattering angle of a probe mass around a Schwarzschild black hole).

We have chosen to drop this statement altogether. We could have stated that elliptic integrals appear only when computing the circumference of an elliptic orbit, but decided this to not be essential here.

4.2) page 1 the text writes: "The first non-trivial order for BH scattering (G^2) was found in 1979²⁵" where Ref. 25 refers to a 1979 paper of Westpfahl+Goller.

Actually, the latter paper does not compute the scattering of two BHs, but rather gives the 2PM equations of motion.

The first computation of the G^2 scattering was published by {Westpfahl:1985ts}.

We have updated the reference accordingly.

4.3) I am not happy with Fig. 2 and the corresponding caption and text.

One of the conceptual beauties of the present work is the appearance of CY periods. I encourage the authors to explain in a more pedagogical manner the precise meaning, and progression in complexity, between elliptic integrals, K3 periods and CY3 periods. E.g. the authors could start recalling the formulas for the usual (Legendre-form) complete elliptic integrals

$\int_a^b dx/y$ based on a cubic curve

$$y^2 = x(x-1)(x-z)$$

with an integral taken either from 0 to 1 or 1 to z (or whatever).

Then Fig. 1 could feature the Legendre cubic curve (as a real planar curve) and the text explain that the integral of the (holomorphic) 1-form $\omega_1 = dx/y$ on independent cycles (when viewing the curve, a la Riemann as a real 2-surface) gets generalized to integrals of (holomorphic) 2-forms over K3 surface, and integrals of (holomorphic) 3-forms of CY3. The current Fig 1 seems to mix real (n-dimensional) sections of complex varieties (of complex dimension n) with (for n=1) 2n-dimensional real representations. I find this more confusing than clarifying.

In addition, in preparation for the K3 and CY3 Picard-Fuchs ODEs, the authors could recall that complete elliptic integrals satisfy a well-known first-order system (first derived by Legendre) leading to a second-order ODE for each elliptic period.

The motivation for the above suggestions is that I think that a general physicist has some knowledge of elliptic integrals (and their link to an holomorphic 1-form dx/y) so that such pedagogical reminders will help many readers to better grasp the meaning of the K3 and CY3 periods entering your results.

We very much welcome these comments of the referee as they completely agree with our point of view and give us the opportunity to state some of the important conceptual points more explicitly. In our first version we tried to limit the number of equations and mathematical explanations.

Of course these limitations are not gone, but we found the following compromise:

We added very shortly and elementarily the case of the Legendre curve, its algebraic definition, its holomorphic one form, the direct relation of its periods to elliptic functions as well as the second order Picard-Fuchs equations the latter fulfill. We also stressed the property of the series of Calabi-Yau n-folds to exhibit in complete analogy a unique holomorphic n-form. This can be found under Fig 2 in the main text. It allowed us to make the capture of Fig 2 more stringent.

We already thought in our first version to restrict ourselves in the case of the elliptic curve to the real projection as we do for the K3, which exhibits certain aspects more clearly as the referee points out. The problem is that we cannot apply the same logic of a real projection to the CY 3-fold case, whose occurrence is conceptually most significant. Our solution of this case is a projection with a 4-d kernel, which reflects its $Z/(8Z) \times Z/(4Z)$ symmetry. The projection can be chosen such that the real 3-cycles appear one real dimensional, so this aspect is correctly represented. In summary we find Fig 2 and its new capture are an instructive compromise that hopefully reveals "... the conceptual beauties of the present work..." better (see also the updates to 4.5)).

4.4) p 4 It would be good to write "In the velocity-odd" sector of integrals (instead of "in the odd sector")

Agreed, this has now been updated. We say "parity odd" to be consistent with previous nomenclature.

4.5) p 4 near Eq (5). As the appearance of a CY3 is conceptually novel in GW physics, I would encourage the authors to describe in more details the algebraic equations defining their CY3. Can one write some explicit equation similar to a quintic (containing x as unique parameter in its coefficients) in CP^4 , together with an explicit form of the (holomorphic) 3-form ω_3 ?

Then, after writing the Picard-Fuchs equation the authors might mention that one solution of this equation is a (rather simple) generalized hypergeometric function ${}_pF_q[a, \dots, b, \dots, x]$.

I asked Mathematical DSolve to solve eq (5) and immediately got concrete solutions. Notably

$(-1)^{1/4} x C[1] \text{HypergeometricPFQ}[\{1/2, 1/2, 1/2, 1/2\}, \{1, 1, 1\}, x^4]$.

Indeed, I think that it would be good

for the reader to have in mind some explicit expression

(in ${}_pF_q$ form) for at least one of the periods, so that (s)he has a better

feeling for the complexity of later using such a function inside an iterated integral.

Again we fully agree with the referee and suggest in view of the space limitation the following compromise: In the text we highlighted the fact that (5) is the Picard-Fuchs equation for the periods over the holomorphic 3-form over the four real three dimensional cycles of the CY 3-fold and therefore the analogy to the structures reviewed before for the Legendre curve should be obvious.

The aspect of the one parameter hypergeometric rank four motives was an important point in the detailed study of the latter by one the authors with Boehnisch, Scheidegger and Zagier "D-brane masses at special fibres of hypergeometric families of Calabi-Yau threefolds, modular forms, and periods" (Ref 72) where two explicit forms of the equations of our particular CY 3-fold as well as the corresponding representations for the holomorphic 3-form were given.

The difficulty is that in order to understand the story of the one parameter Calabi-Yau 3-fold and its homology and cohomology thoroughly one has to introduce mirror symmetry, i.e. the quotient of its algebraic representation in CP^7 by those symmetries now mentioned in Fig 2. In shortening the latter consideration one runs high risks of creating severe mathematical imprecisions. Therefore we mention the fact that the PF operator is hypergeometric and refer for the geometrical background to the Ref 72 as well as to previous work Ref 61. Since we already introduced the Legendre curve, we also mention the intriguing fact that the CY 3-fold and the K3 we encounter can be related to the Legendre curve, by well known mathematical procedures. We do all this in a footnote for space reasons.

We have the feeling that the strength of our work is its strong interdisciplinary nature that leads to truly new techniques with wide applications leading here to specific new physical results. Since the focus is on the latter it seems for now a footnote in the main text to this particular mathematical background with precise referencing is adequate. If the editor encourages us to expand, we are more than happy to do so.

More details are then provided in the Methods section (footnote 5 on page 7): The possibility of using a generalized hypergeometric function ${}_pF_q$ to represent the solution to Eq. (5) is now discussed.

4.6) p 4 below eq 6 "rescaled energy" → "mass-rescaled energy"

We thank the referee for this, it is now updated.

4.7) p 5 Fig 6 and its caption:

I do not understand how Fig 6 was made. First, the caption says that it uses an NR simulation from Ref 63 with $v=0.5$. But, I do not see any such simulation in Ref. 63. The closest relative velocity would be for the case that $E_{in} = 1.04033$ in Table I, corresponding to $v=0.5125$ (which is significantly different from 0.5). In addition, the corresponding rescaled impact parameters b/GM are 16,14,12,10,9.6,9.5,9.4,9,8.8. The corresponding scattering angles seem to be those plotted in Fig. 6, but I do not understand why the impact parameters b/GM used in Fig 6 are not those listed above. They seem to differ by some factor?? Am I saying something wrong? Anyway, the reader needs more explanations to understand how Fig. 6 was made.

Indeed, the NR points in Fig. 6 corresponds to the NR data points of [69] with velocity $v=0.5125$ (that is $E=1.04033$). We had on grounds of brevity only displayed one significant digits but have now added the missing digits such that the captions now reads $v=0.5125$. As to the impact parameters: For each simulation, Ref. [69] reports three quantities characterizing them: b_{NR} , E_{in} and L_{in} . It is correct that the values of b_{NR} are 16, 14, 12 etc. The values that we use for the impact parameter, however, are the ones inferred from E_{in} and L_{in} by expressing the impact parameter in terms of the total energy and total angular momentum and they differ somewhat from b_{NR} . Our impression is that while b_{NR} gives a rough estimate of the impact parameter it is rather an input value to the NR simulations defined at a certain initial distance ($100 GM/c^2$). Instead E_{in} and L_{in} reflect the initial state as precisely as possible and we therefore use these to infer the initial value of the impact parameter.

We could offer to include a footnote explaining this subtle point.

4.8) p 5 Fig. 7 and its caption.

I do not understand what is plotted in Fig. 7.

What is this rescaling by "the leading non relativistic value up to the G^5 " ???

I would like to make a suggestion to clarify the physical content of the new results obtained by the authors concerning the G^5 radiated energy-momentum.

Indeed, as the authors mention in their Eq. 24,

the full content of the G^5 radiated energy-momentum is described by four functions of γ , which they denote $r1(\gamma)$, $r2(\gamma)$, $r3(\gamma)$ and $r4(\gamma)$ (the latter being at the 2SF level).

[Let me mention in passing that this was explicitly pointed out in Ref. 95 (=BDG2023), see Eq 11.8 of [95], involving the functions

$p_{[2/2]^{1+2,G^5}}(\gamma, \nu)$ which is linear in ν ,
 $p_{[1/2]^{1-2,G^5}}(\gamma)$, and $p_{[1/2]^{b12,G^5}}(\gamma)$.]

I would then suggest that the authors replace (or complete?) Fig 7 by a Figure displaying the three building blocks they determined, namely $r1(\gamma)$, $r2(\gamma)$, and $r3(\gamma)$ as functions of γ (or v).

[Note also that Eq, 11.28 of [95] has displayed the known PN knowledge of these functions.]

Independently of the issue of displaying the results of the authors in some figure, I expected the authors to discuss the high-energy behavior of their E_{rad} results. Does it worsen the issue found at G^3 and G^4 that the perturbative E_{rad} exceeds (at high γ 's) the incoming energy even when considering a small, fixed scattering angle?

This figure 7 has now been removed as the normalization was very subtle.

The reference BDG2023 [104] has now been added above Eq. (24). We prefer not to get into too much detail with the individual functions $r_1/r_2/r_3$ in the main text, though, as our space is limited.

Indeed, \mathcal{E} continues to diverge in the high energy limit as was observed at G^4 .

4.9) p 5 In eq (8) should there not be a factor $1/2$??

Yes, that was a typo; it has now been fixed.

4.10) p 9 I would like to remind the authors that the definition of the relative scattering angle (with Eq. 25) was introduced in Ref 94 (= BDG2021)

This citation has now been added.

4.11) p 9 second column: The authors mention that they have checked that their G^5 scattering results agree with PN results up to the 5PN order. Let me mention that more PN terms in the (radiation-reacted) scattering have been derived. In particular, results in the PN literature yield a *complete* answer for the radiation-reacted scattering *at the 5.5PN level* (i.e. one power of v beyond the 5PN level).

Indeed, Ref. Bini:2020hmy} D.~Bini, T.~Damour and A.~Geralico, "Sixth post-Newtonian nonlocal-in-time dynamics of binary systems," Phys. Rev. D **102**, no.8, 084047 (2020)

has derived the 5.5PN *conservative* contribution to the relative scattering angle. Interestingly, this 5.5PN contribution is due to subtle tail-of-tail (or tail²) nonlocal interactions; see Eq. 5.10 there. This conservative contribution to scattering can then be completed by corresponding radiation-reaction contributions

which have been derived in [94] and [95] (BDG2021 and BDG2023).

See notably Table IX of [94] and Table II of [95], as well as Sections 12 I and 12 J of [95] which also completed the linear-response scattering formula for the (relative) scattering angle, by deriving the fully nonlinear radiation-reaction contributions to the G^5 impulse modulo a 2SF contribution in the longitudinal component of the impulse.

In particular, I have checked that by combining the above-cited PN results one exactly reproduces the value of the

1SF relative scattering angle, i.e. modulo a rescaling, the quantity $\theta^{(5,1)}$ (Eq S.3a of the Supplementary

information), up to the 5.5PN level (i.e. up to the v^1 term).

[By contrast, the 6PN-level results derived in the above references do not yield (even at 1SF) a complete

radiation-reacted result to be compared with the 1SF G^5 results presented by the authors.]

I encourage the authors to redo this 5.5PN check (which brings a further deep confirmation of their results, in

view of the many tail, and tail², effects included in the needed PN results), or at least to mention that complete

5.5PN results exist in the literature.

We thank the referee for bringing this very important point to our attention and sincerely thank him for the detailed and non-trivial independent check of our results. We have now also checked that our scattering angle at 5.5PN order and confirm agreement with the result of Bini, Damour, and Geralico.

4.12) SUPPLEMENTARY INFORMATION

Eqs S.3, see above concerning the 5.5PN check of the scattering

We have clarified that the check is now performed up to v^1 rather than v^0 as before.

Referee #4 (Remarks to the Author):

The current paper uses effective field theory to describe the scattering process of two gravitationally interacted particles. Along the EFT framework, the authors expand the solution as a series respect to the gravitational constant G . With 'high-precision', the authors means they did calculation till G^5 . It is interesting and important to get such high order result. In addition, the authors find that the results show an interesting special function which is related to CY manifold. The appearance of such a special function give us insight of the character of gravity. These results are interesting and important enough for publication in Nature.

We thank the referee for this warm review of our work!

As a Nature paper, I have several comments and suggestions in the following.

1. If the two particles mean matter such as neutron star, there is direct meaning of mass (m_1, m_2). But if pure black holes are involved, the masses are reduced concepts ($R/2G$) where R is the characteristic size respect to the spacetime curvature. Consequently the action can be factorized as $1/G$ and another G independent expression. This leads us to Einstein equation without G . In this situation, the G series do not make sense. In fact the expansion is essentially respect to R/d where d is the separation of the two objects. Then conceptually I suggest the authors explain the difference between EFT and usual PN approximations.

The referee is absolutely correct that the PM expansion parameter for the scattering problem is, strictly speaking not G but rather GM/b where M is the total mass and b is the impact parameter. We have clarified this by adding a sentence above (1). Still, the use of the powers of G is common in the PM community and, to our mind also of pedagogical value. As to explain the difference between our WQFT and the usual PN approximation, we have introduced the latter in the introductory paragraphs, making it clear that PN is a non-relativistic and weak gravity limit, whereas PM only assumes a weak gravity situation being valid for arbitrary velocities. All changes to the manuscript are typeset in red.

2. For the faced problem, it is a standard classical problem. In this sense the action (1) is exact. There is no effective meaning at all. I just guess the authors call the theoretical framework EFT just because they treat it as a quantum field theory. And consequently the Feynman diagram technique is applicable. Then I have a question. What kind of rules guarantee the quantum field treatment can result in the solution of the original standard classical problem? For comparison, we have EOBNR models for CBC systems. EOBNR theory also uses quantum field theory technique. But we have to notice that EOB just gives better convergent result than PN theory. Theoretically we are not sure EOB results corresponds to the real solution of Einstein equation. Only after the calibration by numerical relativity, we are sure the EOB results is trustable. Regarding the EFT involved in the current paper, I suggest the authors give some comments on this point.

We thank the referee for raising this crucial point that was unintentionally skipped in the discussion of WQFT. The key principle being exploited in our approach is that *tree-level one-point functions*, given by a sum of diagrams with a single outgoing line and no internally closed loops, *solve the classical equations of motion*. This was shown in a classic QFT paper by Boulware and Brown in 1968. We added this statement on page 3 in the discussion of the WQFT formalism.

3. There is one more question about gauge problem in general relativity. Along with the EFT, the authors work in some coordinate where the spacetime metric takes form $g_{\mu\nu}$. I imagine that initially ($t \rightarrow -\infty$) $g_{\mu\nu}$ is Minkowsky. My concern is final ($t \rightarrow +\infty$) state of $g_{\mu\nu}$. If final $g_{\mu\nu}$ is Minkowsky, the meaning of angles is straightforward and the presentation of the current paper is without problem. If it is not Minkowsky, the metric should affect the geometric meaning. Here with 'Minkowsky' I mean the metric form of Minkowsky under the

inertial coordinates, $\eta_{\mu\nu}$. In general, the gravitational memory may lead the final metric do not equal to $\eta_{\mu\nu}$ although we are sure the final spacetime is Minkowsky. Since the accuracy of the current paper is extremely high, how about this effect? Or to say the memory effect is higher than G^5 order?

This is a very good and subtle point. The gravitational memory effect is of order GM/r (where r is the distance to the detector), see e.g., arxiv.org 2101.12688 [gr-qc] eq (11) and (28) computed in our formalism. Here, we compute scattering observables in a GM/b expansion, where b is the impact parameter. As $r \gg b$, the memory may be neglected here.

4. As the authors said, the CY n-folds "generalize the elliptic integrals and are known to arise in perturbative QFT in multiloop Feynman integrals". Does this mean the CY n-folds appear for any perturbative QFT instead of specifically for the EFT of GR?

CY n-folds are indeed known to appear in high-loop Feynman integrals in perturbative quantum field theory. Examples are the Banana and Ice Cone Families or the electron and photon self-energy in QED. As such they are generically expected to appear at high enough order, yet they have so far not appeared in a final *observable*. The reason being that in particle physics scattering amplitudes are generically significantly simpler than individual Feynman integrals. Our work is the first to demonstrate the appearance of a CY3 period in a physical observable.

5. The authors emphasized that "Until now, no physical observables in elementary particle physics or gravity have been reported where CY3 periods appear." Then does the result shown in the current paper mean we can use binary black hole observations to relate CY3?

Indeed, that is the case for binary black hole scattering. The CY3 period enters in the radiated energy and three-momentum recoil at the 5PM-1SF order, as we have shown. In these observables, they contribute to the tail-of-tail effect, i.e., the backscattering of radiative gravitons off the gravitational potential in the space-time bulk. In that sense one may attribute their appearance to a physical effect. We have included a passage in the final discussion on the last page of the main text, discussing this point.

Referee #1 (Remarks to the Author):

I have carefully read the revised manuscript as well as the authors' responses. As a referee, I also gained a lot from this process, and would like to express my gratitude to the authors and the other anonymous referees. Below are my further comments:

1. I fully agree with and appreciate the authors' strategy of presenting the material in a "technically graded" manner. However, to maximize the accessibility of the paper, it is essential to provide clear and intuitive explanations of key mathematical structures—such as elliptic curves, K3 surfaces, and Calabi-Yau manifolds—particularly in the introductory part of the paper.

We thank the referee for their valuable work in the review process and are happy to hear that they profited from the process. We have now improved the exposition of mathematical structures (elliptic curves, CY manifolds and periods) in the introductory part of the main text, cutting back on technical terms as far as possible in order to increase the accessibility of the paper. This effort mirrors request of referee #4 and the editor.

2. Following my earlier suggestion, the authors have cited Goldberger and Rothstein's seminal work on the worldline EFT approach in their revised version.

3. The authors observed that CY integrals "have so far not appeared in a final observable," but their explanation is somewhat oversimplified. The claim that "the most complicated structures of the related Feynman integrals drop out" in amplitudes is generally valid for relatively simple theories (and processes). In more complex theories, such as the electroweak theory, the presence of multiple scales—such as the masses of Higgs and W/Z bosons—can lead to results that involve elliptic/CY integrals and even more complicated functions. In such cases, observables are often best expressed numerically rather than in terms of huge analytic expressions involving complicated functions. While I am not happy with "...Calabi-Yau manifolds appear for the first time in nature", I do acknowledge that classical two-body scattering, due to the simplicity of its final results, provides an excellent testing ground for exploring the interplay between physics and Calabi-Yau geometry.

We agree with the referee that in complex many scale standard model processes results leading to higher CY periods are in general to be expected. Yet we stand firm to our statement that as of today no observable featuring CY3-fold periods has been reported, neither in elementary particle physics nor in gravitational wave physics. While it could be that existing purely numerical results for standard model processes do indeed contain such functions, this is at present unknown and to our mind does not diminish the novelty of our results. However, we have dropped the controversial statement "Calabi-Yau manifolds appear for the first time in nature" in the manuscript now.

Referee #2 and #5 (Remarks to the Author):

The authors have addressed most of the comments/suggestions in our previous referee report. Overall, their revisions have strengthened the manuscript and improved the clarity of their presentation. In this revised manuscript, the authors have made more connections between the mathematical features of their results and physical observables. For example, the novel Calabi-Yau (CY) 3 periods appearing at the 5PM order could be connected to the repeated back-scattering of radiative gravitons off the spacetime potential. They have also pointed out that although CY periods have been known to appear in individual Feynman integrals for a long time, it is the first time that the CY3 period

appears in the final physical results, which could thus spur more research into understanding the importance of the CY geometry in the scattering physics. Given their response and the revisions, we are satisfied with the technical rigor of this work and agree with the other referees that this work can be important in studying gravitational scattering.

We appreciate the significant contributions this work has made to the gravitational two-body scattering problem. As the authors highlight in their response, these contributions may have broader implications for studying scattering amplitudes in other field theories, such as quantum electrodynamics. That said, we still share the concern raised by Referee 1 that “due to the specialist nature of perturbative quantum field theory, some parts of the presentation may be overly technical, potentially limiting accessibility for a significant portion of Nature’s readership.” While this work is undoubtedly valuable to the scattering amplitude community, its relevance to the broader gravitational physics community is moderate. For example, even within the community of gravitational wave modeling, it is unclear how the sophisticated techniques developed here, specifically for scattering problems, inform or help other key analytical or semi-analytical approaches, such as gravitational self-force and black hole perturbation theory.

In addition, we share Referee 1’s concern, also mentioned in our previous report, regarding whether the “considerable emphasis on the appearance of Calabi-Yau manifolds” is necessary or appropriate. Despite the authors’ explanation to Referee 1, it seems that lower-order CY periods have already appeared in the physical observables of lower-order PM calculations, for example, the K3 period at $O(G^4)$. Given this, the appearance of the CY3 period at $O(G^5)$ may not be that surprising, and perhaps it is an expected outcome of performing higher-order PM calculations. Therefore, it is unclear how this feature, arising naturally from perturbative expansion and the use of Feynman integrals, advances our fundamental understanding of gravity itself.

In total, we are concerned that this work may not fully align with Nature’s emphasis on “interdisciplinary interest.” We believe a more specialized journal is a better fit because it would ensure this important, technical paper reaches a more appropriate readership.

We thank the referees for their continued work and are pleased that they deem our revised version to have further strengthened our manuscript and improved the clarity of its presentation. We would like to comment on the impact of our results and techniques on another key approach to the gravitational two-body problem, the gravitational self-force program. In fact, this semi-analytical approach (being complementary to the PM expansion) has been quite influenced by the QFT driven advances in the PM expansion of which our present work sets a new landmark. Leading self-force researchers such as Leon Barack, Adam Pound or Chris Kavanaugh are now studying the scattering problem with their methods. One reason being that e.g. the subtle frame and gauge dependencies in the bound system are gone for the asymptotic problem, presenting a clean probe. As such, first cross community publications, e.g. Barack, Bern et al (PhysRevD.108.024025), and joint conferences (1st and 2nd Workshop on “Self Force and Amplitudes” <https://indico.global/event/4539/> in Edinburgh (2024) and this year in Southampton) have emerged. There has been quite some interest of the self-force community on our advances that we witnessed in conferences and discussions in the past. In fact, we believe that a cross-fertilization of techniques developed in these two traditionally far separated communities bears the potential of further key advances in our understanding of the gravitational two-body problem. In addition, the fact that advanced mathematical structures as reported on in our paper, traditionally studied in pure math and mathematical physics have been shown to appear in the radiated

energy of a black hole encounter in our universe does carry sufficient interdisciplinary interest to be reported in Nature.

Referee #3 (Remarks to the Author):

All my suggestions were adequately addressed (sometimes with an understandable constraint coming from page limits) in this revised version. I have also looked at the modifications aimed at addressing the comments of the other referees.

I think these modifications satisfactorily addressed the comments of the other referees, and that the paper is now improved, and will be understandable, and of interest, to a large readership. I warmly recommend publication of this landmark paper as is.

We are more than grateful to the referee for his extremely careful review work, the very useful comments that improved our manuscript and the praising of our work.

Referee #4 (Remarks to the Author):

Thanks the authors for their careful checking of my concerns. After reading the revised paper I have one more question. Since the period of CY manifolds appeared in GR scattering is the most important finding of the current paper, I suggest the authors to explain these quantities in a simpler way.

We have improved the introduction of the mathematical concepts leading to CY periods in the introductory part, thereby also reacting to comments of referee #1 and the editor.

CY manifolds can be definitely scaled. So I guess the specific periods are not fixed numbers. Instead some ratio may be a geometric quantity. For an example, torius admits π which corresponds to the period of toriuses. I guess K3 and CY3 may admit some similar quantities related to the periods mentioned by the authors. But maybe I am wrong. Anyhow, if the authors can explain the periods in this kind of simpler way, I believe our readers can benefit more from the paper.

Indeed, the CY3 as the K3 periods depend on the parameter x related to the relativistic gamma factor of the initial relative velocity. One may think of them as parametrizing the shapes of these manifolds. We mentioned this in the caption of fig. 2: “[the periods] depend on the single modulus x of the CYs”. In order to improve the presentation in this point, we have changed this to: “[the periods] depend on the so-called modulus x (related to the relative velocity of the black holes $v_1 \cdot v_2 / c^2 = (x + 1/x)/2$) parametrizing the shape of CYs” in the caption.

In conclusion, I am happy to recommend this paper for publication in Nature because it's interesting and important enough.

We thank the referee for their work and assessment!

Version 0 Referee #2 and #5 Report

This work presents the first complete calculation of scattering observables for binary compact objects, modeled as point particles, in the post-Minkowskian (PM) expansion up to the fifth order in the gravitational constant G and the second order in the symmetric mass ratio ν . The authors compute the change in four-momentum, the scattering angle, and the energy radiated by gravitational waves (GWs) up to $\mathcal{O}(G^5)$ and $\mathcal{O}(\nu^2)$, achieving agreement with lower-order PM results, post-Newtonian (PN) expansions, and numerical relativity simulations. While focused on unbounded orbits, their results can potentially provide valuable input to other endeavors, such as the effective-one-body formalism, for modeling the GWs emitted by binary inspirals (with bounded orbits), which are the main targets of LVK detectors. Novel mathematical features, such as high-fold Calabi-Yau (CY) periods, also emerge in their observables. These new geometrical quantities may reveal interesting connections between classical scattering problems and geometries underlying quantum gravity theories, such as string theory.

This work is technically sound and has practical importance for developing efficient and high-precision waveforms required by next-generation GW detectors. This work focuses on employing cutting-edge techniques, such as integration-by-parts reduction, to tackle the computational challenges of evaluating a vast number of Feynman integrals. These techniques, while impressive, also raise questions of whether this work is of enough interest to a broader community. Many of these techniques seem to have mature packages or algorithms developed for more general problems in differential equations. While the authors have made novel improvements, such as methods for decoupling differential equation sectors with different CY periods, the practical utility of these advances outside gravitational scattering remains unclear. Furthermore, all these technical details also hide what new physics the authors have learned or could potentially learn by conducting these high-order PM calculations besides obtaining more precise waveforms. For example, this work emphasizes the novel appearance of the CY3 periods in their observables. However, whether different CY periods could be related to specific observables in GW detections is unclear. If they could, how would they *concretely* further our understanding of quantum gravity due to their connections to string theory?

For these reasons, this work may not be suitable for publication in Nature. Despite these concerns, the results in this work are significant, and the manuscript is well-written so that it could get published in, for example, other

journals with a more specific focus.

Below we include more specific, minor comments/suggestions on the manuscript:

1. In the second and third paragraphs of page 1, the authors briefly review various methods for waveform modeling and argue analytical methods can help reduce the high computational expense of numerical relativity. While the progress in PM formalism has been discussed in detail, the authors could also add more discussion on the current stages of other analytical or semi-analytical approaches, such as black hole perturbation theory, gravitational self-force, and PN formalism. This will help the readers understand where this work sits in the broader efforts of waveform modeling.
2. At the end of page 1, the authors mentioned that at least the $\mathcal{O}(G^5)$ precision will be needed for GW detectors. Could the authors also comment on what PM order will be enough for next-generation detectors? Suppose $\mathcal{O}(G^5)$ is not even sufficient. How hard will it be to extend this calculation to higher PM orders, especially compared to other efforts to reduce the computational cost of waveform modeling, such as developing more efficient numerical relativity algorithms or surrogate models?
3. On page 2, the authors discuss that the K3 periods have been encountered at $\mathcal{O}(G^4)$, and the CY3 period appear when the calculations are extended to $\mathcal{O}(G^5)$, as in this work. In this case, shall one expect higher-fold CY periods to appear for calculations at a higher PM order? For $\mathcal{O}(G^n)$ corrections, shall one always expect $(n - 2)$ -fold CY period to appear? Or could the structure differ?
4. As mentioned above, are these additional high-fold CY periods related to new physical features when improving the description of scattering encounters? If they are, some explanation of the correspondence of the first few CY periods to physical observables might help the readers have a clearer physical picture.
5. In Fig. 7, it seems that the additional radiated energy due to $\mathcal{O}(G^5)$ corrections is comparable to the total energy radiated up to $\mathcal{O}(G^3)$. Since PM formalism is a perturbative approach, should one expect the correction to the radiated energy to decrease generally at a higher PM order, so that the radial energy converges?
6. Could the authors more clearly define the symbols I_1, \dots, I_3 above Eq. (13)? If they are defined in the earlier part of the paper, the authors may consider briefly reviewing the definition here.
7. In Eq. (13), the symbol θ is defined to be $x \frac{d}{dx}$, while it was also used as the scattering angle in other parts of this paper. Could the authors choose a different symbol for $x \frac{d}{dx}$ to avoid confusion?

8. Could the authors clarify the meaning of “lower $G_i(x)$ ” functions above Eq. (17)? Does it mean that, for example, the differential equation of $G_4(x)$ will depend on $G_1(x), \dots, G_3(x)$?

Version 1 Referee #2 and #5 Report

The authors have addressed most of the comments/suggestions in our previous referee report. Overall, their revisions have strengthened the manuscript and improved the clarity of their presentation. In this revised manuscript, the authors have made more connections between the mathematical features of their results and physical observables. For example, the novel Calabi-Yau (CY) 3 periods appearing at the 5PM order could be connected to the repeated back-scattering of radiative gravitons off the spacetime potential. They have also pointed out that although CY periods have been known to appear in individual Feynman integrals for a long time, it is the first time that the CY3 period appears in the final physical results, which could thus spur more research into understanding the importance of the CY geometry in the scattering physics. Given their response and the revisions, we are satisfied with the technical rigor of this work and agree with the other referees that this work can be important in studying gravitational scattering.

We appreciate the significant contributions this work has made to the gravitational two-body scattering problem. As the authors highlight in their response, these contributions may have broader implications for studying scattering amplitudes in other field theories, such as quantum electrodynamics. That said, we still share the concern raised by Referee 1 that “due to the specialist nature of perturbative quantum field theory, some parts of the presentation may be overly technical, potentially limiting accessibility for a significant portion of Nature’s readership.” While this work is undoubtedly valuable to the scattering amplitude community, its relevance to the broader gravitational physics community is moderate. For example, even within the community of gravitational wave modeling, it is unclear how the sophisticated techniques developed here, specifically for scattering problems, inform or help other key analytical or semi-analytical approaches, such as gravitational self-force and black hole perturbation theory.

In addition, we share Referee 1’s concern, also mentioned in our previous report, regarding whether the “considerable emphasis on the appearance of Calabi-Yau manifolds” is necessary or appropriate. Despite the authors’ explanation to Referee 1, it seems that lower-order CY periods have already appeared in the physical observables of lower-order PM calculations, for example, the K3 period at $\mathcal{O}(G^4)$. Given this, the appearance of the CY3 period at $\mathcal{O}(G^5)$ may not be that surprising, and perhaps it is an expected outcome of performing higher-order PM calculations. Therefore, it is unclear how this feature, aris-

ing naturally from perturbative expansion and the use of Feynman integrals, advances our fundamental understanding of gravity itself.

In total, we are concerned that this work may not fully align with Nature's emphasis on "interdisciplinary interest." We believe a more specialized journal is a better fit because it would ensure this important, technical paper reaches a more appropriate readership.